# V-DETR: DETR with Vertex Relative Position Encoding for 3D Object Detection

**Yichao Shen[1]\*, Zigang Geng[2]\*, Yuhui Yuan[3]\*[†], Yutong Lin[1], Ze Liu[2], Chunyu Wang[3], Han Hu[3], Nanning Zheng[1][†], Baining Guo[3]**

[1]National Key Laboratory of Human-Machine Hybrid Augmented Intelligence, National Engineering Research Center for Visual Informationand Applications, and Institute of Artificial Intelligence and Robotics, Xi'an Jiaotong University
[2]University of Science and Technology of China
[3]Microsoft Research Asia

## Abstract

We introduce a highly performant 3D object detector for point clouds using the DETR framework. The prior attempts all end up with suboptimal results because they fail to learn accurate inductive biases from the limited scale of training data. In particular, the queries often attend to points that are far away from the target objects, violating the locality principle in object detection. To address the limitation, we introduce a novel 3D Vertex Relative Position Encoding (3DV-RPE) method which computes position encoding for each point based on its relative position to the 3D boxes predicted by the queries in each decoder layer, thus providing clear information to guide the model to focus on points near the objects, in accordance with the principle of locality. Furthermore, we have systematically refined our pipeline, including data normalization, to better align with the task requirements. Our approach demonstrates remarkable performance on the demanding ScanNetV2 benchmark, showcasing substantial enhancements over the prior state-of-the-art CAGroup3D. Specifically, we achieve an increase in $AP_{25}$ from 75.1% to 77.8% and in $AP_{50}$ from 61.3% to 66.0%.

## 1 Introduction

3D object detection from point clouds is a challenging task that involves identifying and localizing the objects of interest present in a 3D space. This space is represented using a collection of data points that have been gleaned from the surfaces of all accessible objects and background in the scene. The task has significant implications for various industries, including augmented reality, gaming, robotics, and autonomous driving.

Transformers have made remarkable advancement in 2D object detection, serving as both powerful backbones (Vaswani et al., 2017; Liu et al., 2021a) and detection architectures (Carion et al., 2020). However, their performance in 3D detection (Misra et al., 2021) is significantly worse than the state-of-the-art methods. Our in-depth evaluation of 3DETR (Misra et al., 2021) revealed

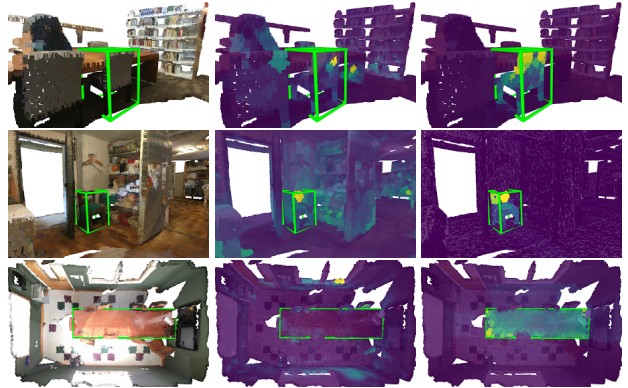

Figure 1: The left column shows the 3D scans from the ScanNetV2 in the rear/front/top-down view. We display one of the ground-truth bounding boxes with a green 3D box. The middle column shows the decoder cross-attention map based on plain DETR. Attention weights are distributed over many positions even outside the ground-truth box. The right column shows the decoder cross-attention map based on plain DETR + 3DV-RPE. Attention weights focus on the sparse object boundaries of the object located in the ground-truth bounding box. The color indicates the attention values: yellow for high and blue for low.

that the queries often attend to points that are far away from the target objects (Figure 1 (b) shows three typical visualizations), which violates the principle of *locality* in object detection. The principle of locality dictates that object detection should only consider subregions of data that contain

---

\*Core contribution.
[†]Corresponding Author.

the object of interest and not the entire space. Besides, the behavior is also in contrast with the success that Transformers have achieved in 2D detection, where they have been able to effectively learn the inductive biases, including locality. We attribute the discrepancy to the limited scale of training data available for 3D object detection, making it difficult for Transformers to acquire the correct inductive biases.

In this paper, we present a simple yet highly performant method for 3D object detection using the transformer architecture DETR (Carion et al., 2020). To improve locality in the cross-attention mechanism, we introduce a novel 3D Vertex Relative Position Encoding (3DV-RPE) method. It computes a position encoding for each point based on its relative offsets to the vertices of the predicted 3D boxes associated with the queries, providing clear positional information such as whether each point is inside the boxes. This information can be utilized by the model to guide cross-attention to focus on points inside the box, in accordance with the principle of locality. The prediction of these boxes is consistently refined as the decoder layers progress, resulting in increasingly accurate position encoding.

To mitigate the impact of object rotation, we propose to compute 3DV-RPE in a canonical object space where all objects are consistently rotated. Particularly, for each query, we predict a rotated 3D box and compute the relative offsets between the 3D points rotated in the same way, and the eight vertices of the box. This results in consistent position encoding for different instances of the same object regardless of their positions or orientations in the space, greatly facilitating the learning of the locality property in cross-attention even from limited training data. Figure 1 (c) visualizes the attention weights obtained by our method. We can see that the query for detecting the chair nicely attends to the points on the chair. Our experiment demonstrates that 3DV-RPE boosts the performance.

We also systematically enhance our pipeline from various aspects such as data normalization and network architectures based on our understanding of the task. For example, we propose object-based normalization, instead of the scene-based one used by the DETR series, to parameterize the 3D boxes. This is because the former is more stable for point clouds which differs from 2D detection where the sizes of the same object in images can be very different depending on the camera parameters, impelling them to use image size to coarsely normalize the boxes. Besides, we also evaluate and adapt some of the recent advancement in 2D DETR.

We conduct thorough experiments to empirically show that our simple DETR-based approach significantly outperforms the previous state-of-the-art fully convolutional 3D detection methods, which helps to accelerate the convergence of the detection head architecture design for 2D and 3D detection tasks. We report the results of our approach on two challenging indoor 3D object detection benchmarks including ScanNetV2 and SUN RGB-D. Overall, compared to the DETR baseline (Misra et al., 2021), our method with 3DV-RPE improves $AP_{25}/AP_{50}$ from $65.0\%/47.0\%$ to $77.8\%/66.0\%$, respectively, and reduces the training epochs by 50%. Particularly, on ScanNetV2, our approach outperforms the very recent state-of-the-art CAGroup3D (Wang et al., 2022a) by $+2.7\%/+4.7\%$ measured by $AP_{25}/AP_{50}$, respectively.

## 2 RELATED WORK

**DETR-based Object Detection.** DETR (Carion et al., 2020) is a groundbreaking work that applies transformers (Vaswani et al., 2017) to 2D object detection, eliminating many hand-designed components such as non-maximum suppression (Neubeck & Van Gool, 2006) or anchor boxes (Girshick, 2015; Ren et al., 2015; Lin et al., 2017; Liu et al., 2016). Many extensions of DETR have been proposed (Meng et al., 2021; Gao et al., 2021; Dai et al., 2021; Wang et al., 2021; Jia et al., 2022; Zhang et al., 2022), such as Deformable-DETR (Zhu et al., 2020), which uses multi-scale deformable attention to focus on key sampling points and improve performance on small objects. DAB-DETR (Liu et al., 2022) introduces a novel query formulation to enhance detection accuracy. Some recent works (Li et al., 2022; Zhang et al., 2022; Jia et al., 2022; Chen et al., 2022) achieve state-of-the-art results on object detection by using query denoising or one-to-many matching schemes, which addressed the training inefficiency of one-to-one matching. $\mathcal{H}$-DETR (Jia et al., 2022) shows that one-to-many matching can also speed up convergence on 3D object detection tasks. Following the DETR-based approach, GroupFree (Liu et al., 2021b) and 3DETR (Misra et al., 2021) built strong 3D object detection systems for indoor scenes. However, they are still inferior to other methods such as CAGroup3D (Wang et al., 2022a). In this work, we propose several critical modifications

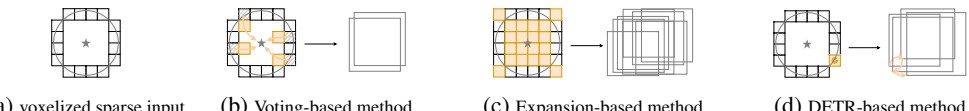

(a) voxelized sparse input  (b) Voting-based method  (c) Expansion-based method  (d) DETR-based method

Figure 2: (a) A simplified sparse 3D voxel space from a top-down perspective. The curve shows the input surface and the small cubes show the voxelized input. The gray five-pointed star ★shows the object's center. (b) The voting scheme estimates offsets for each voxel and we color the voxels (nearer to the object's center) with yellow after voting. The dashed small cubes show the empty space after voting. (c) The generative sparse decoder (GSD) scheme enlarges the voxels around the surfaces, thus creating new voxels both inside and outside of the object (marked with yellow cubes). (d) The DETR-based approach simply selects a small set of voxels (marked with yellow cubes) as the initial object query and iteratively predicts the boxes by refining (marked with the open yellow circles) the object query with multiple Transformer decoder layers. We follow the DETR-based path in this work.

to improve the DETR-based methods and achieve new records on two indoor 3D object detection tasks.

**3D Indoor Object Detection.** We revisit the existing indoor 3D object detection methods that directly use raw point clouds to detect 3D boxes. We categorize them into three types based on their strategies: (i) *Voting-based methods*, such as VoteNet (Qi et al., 2019), MLCVNet (Xie et al., 2020) and H3DNet (Zhang et al., 2020), use a voting mechanism to shift the surface points toward the object centers and group them into object candidates. (ii) *Expansion-based methods*, such as GSDN(Gwak et al., 2020), FCAF3D(Rukhovich et al., 2022), and CAGroup3D(Wang et al., 2022a), which generate virtual center features from surface features using a generative sparse decoder and predict high-quality 3D region proposals. (iii) *DETR-based methods*, unlike these two types that require modifying the original geometry structure of the input 3D point cloud, we adopt the DETR-based approach (Liu et al., 2021b; Misra et al., 2021) for its simplicity and generalization ability. Our experiments show that DETR has great potential for indoor 3D object detection. We show the differences between above-mentioned methods in Figure 2.

**3D Outdoor Object Detection.** We briefly review some methods for outdoor 3D object detection (Yan et al., 2018; Zhou & Tuzel, 2018; Lang et al., 2019; Yin et al., 2021), which mostly transform 3D points into a bird-eye-view plane and apply 2D object detection techniques. For example, VoxelNet (Zhou & Tuzel, 2018) is a single-stage and end-to-end network that combines feature extraction and bounding box prediction. PointPillars (Lang et al., 2019) uses a 2D convolution neural network to process the flattened pillar features from a Bird's Eye View (BEV). CenterPoint (Yin et al., 2021) first detects centers of objects using a keypoint detector and regresses to other attributes, then refines them using additional point features on the object. However, these methods still suffer from center feature missing issues, which FSD (Fan et al., 2023b) tries to address. We plan to extend our approach to outdoor 3D object detection in the future, which could unify indoor and outdoor 3D detection tasks.

## 3 OUR APPROACH

### 3.1 BASELINE SETUP

**Pipeline.** We build our V-DETR baseline following the previous DETR-based 3D object detection methods (Misra et al., 2021; Liu et al., 2021b). The detailed steps are as follows: given a 3D point cloud $\mathbf{I} \in \mathbb{R}^{N \times 6}$ sampled from a 3D scan of an indoor scene, where the RGB values are in the first 3 dimensions and the position XYZ values are in the last 3 dimensions. We first sample about ∼40K points from the original point cloud that typically has around ∼200K points. Second, we use a feature encoder to process the raw sampled points and compute the point features $\mathbf{F} \in \mathbb{R}^{M \times C}$. Third, we construct a set of 3D object queries $\mathbf{Q} \in \mathbb{R}^{K \times C}$ send them into a plain Transformer decoder to predict a set of 3D bounding boxes $\mathbf{B} \in \mathbb{R}^{K \times D}$. We set $K = 1024$ by default. Figure 3 shows the overall pipeline. We present more details on the encoder architecture design, the 3D object query construction, the Hungarian matching, and loss function formulations as follows.

**Encoder architecture.** We choose two different kinds of encoder architecture for experiments including: (i) a PointNet followed by a shallow Transformer encoder adopted by Misra et al. (2021)

or (ii) a sparse 3D modification of ResNet34 followed by an FPN neck adopted by Rukhovich et al. (2022), where we replace the expensive generative transposed convolution with a simple transposed convolution within the FPN neck.

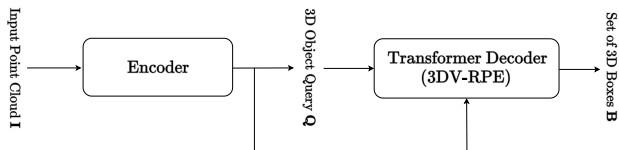

Figure 3: **Illustrating the overall framework of V-DETR for 3D object detection**. We first use an encoder to extract 3D features, and then we use a plain Transformer decoder to estimate the 3D object queries from a set of initialized 3D object queries. In the Transformer decoder multi-head cross-attention layer, we use a 3D vertex relative position encoding scheme for both locality and accurate position modeling.

**3D object query.** We construct the 3D object query by combining two kinds of representations as follows: first, we simply sample a set of K initial center positions over the entire encoder output space and select their representations to initialize a set of 3D content query $\mathbf{Q}_c$. Then we use their XYZ coordinates in the input point cloud space to compute the 3D position query $\mathbf{Q}_p$ with a simple MLP consisting of two linear layers. We build the 3D object query by adding the 3D position query to the 3D content query.

**Hungarian matching and loss function.** We choose the weighted combination of six terms including the bounding box localization regression loss, angular classification and regression loss, and semantic classification loss as the final matching cost functions and training loss functions. We illustrate the mathematical formulations as follows:

$$\mathcal{L}_{\text{DETR}} = -\lambda_1 \text{GIoU}(\hat{\mathbf{b}}, \mathbf{b}) + \lambda_2 \mathcal{L}_{\text{center}}(\hat{\mathbf{c}}, \mathbf{c}) + \lambda_3 \mathcal{L}_{\text{size}}(\hat{\mathbf{s}}, \mathbf{s})$$
$$-\lambda_4 \text{FL}(\hat{\mathbf{p}}[l]) + \lambda_5 \mathcal{L}_{\text{huber}}(\hat{\mathbf{a}}_r, \mathbf{a}_r) + \lambda_6 \text{CE}(\hat{\mathbf{a}}_c, \mathbf{a}_c),$$

where we use $\hat{\mathbf{b}}, \hat{\mathbf{c}}, \hat{\mathbf{s}}, \hat{\mathbf{p}}, \hat{\mathbf{a}}$ (or $\mathbf{b}, \mathbf{c}, \mathbf{s}, l, \mathbf{a}$) to represent the predicted (or ground-truth) bounding box, box center, box size, classification score, and rotation angle respectively, e.g., $l$ represents the ground-truth semantic category of $\mathbf{b}$. CE represents angle classification cross entropy loss and $\mathcal{L}_{\text{huber}}$ represents the residual continuous angle regression loss. FL represents semantic classification focal loss.

**Object-normalized box parameterization.** We propose an object-normalized box reparameterization scheme that differs from the original DETR (Carion et al., 2020), which normalizes the box predictions by the scene scales. We account for one key discrepancy between object size variation in 2D images and 3D point clouds, e.g., a chair's 2D box size may change depending on its distance to the cameras, but its 3D box size should remain consistent as the point cloud captures the real 3D world. In the implementation, we simply reparameterize the prediction target of width and height from the original ground-truth $\mathbf{b}_h$ and $\mathbf{b}_w$ to $\mathbf{b}_h/\hat{\mathbf{b}}_h^{l-1}$ and $\mathbf{b}_w/\hat{\mathbf{b}}_w^{l-1}$, where $\hat{\mathbf{b}}_h^{l-1}$ and $\hat{\mathbf{b}}_w^{l-1}$ represent the coarsely predicted box height and width.

### 3.2 3DV-RPE in Canonical Object Space

Position Encoding (PE) is crucial for enhancing the ability of transformers to comprehend the spatial context of the tokens. The appropriate PE strategy depends on tasks. For 3D object detection, where geometry features are the primary focus, it is essential for PE to encode rich semantic positions for the points, whether they are on/off the 3D shapes of interest.

To that end, we present 3D Vertex Relative Position Encoding (3DV-RPE), a novel solution specifically tailored for 3D object detection within the DETR framework. We modify the global plain Transformer decoder multi-head cross-attention maps as follows:

$$\widehat{\mathbf{A}} = \text{Softmax}(\mathbf{Q}\mathbf{K}^{\text{T}} + \mathbf{R}), \tag{1}$$

where $\mathbf{Q}$ and $\mathbf{K}$ represent the sparse query points and dense key-value points, respectively. $\mathbf{R}$ represents the position encoding computed by our 3DV-RPE that carries accurate position information.

**3DV-RPE.** Our key insight is that, encoding a point by its relative position to the target object, which is coarsely represented by a box, is sufficient for 3D object detection. It is computed as follows:

$$\mathbf{P}_i = \text{MLP}_i(\mathcal{F}(\Delta\mathbf{P}_i)), \tag{2}$$

where $\Delta\mathbf{P}_i \in \mathbb{R}^{\mathsf{K}\times\mathsf{N}\times 3}$ denotes the offsets between the $N$ points and the $i$-th vertex of the $K$ boxes and $\mathbf{P}_i \in \mathbb{R}^{\mathsf{K}\times\mathsf{N}\times h}$ represents the relative position bias term. $h$ is the number of heads. $\mathcal{F}(\cdot)$ is a non-linear function. We will evaluate several alternatives for $\mathcal{F}(\cdot)$ in the experiments. $\mathrm{MLP}_i$ represents an MLP based transformation that first projects the features to a higher dimension space, and then to the output features of dimension $h$. We obtain the final relative position bias term by adding the bias term of the eight vertices, respectively:

$$\mathbf{R} = \sum_{i=1}^{8} \mathbf{P}_i, \tag{3}$$

where $\mathbf{R}$, encodes the relations between the 3D boxes and the points. In the subsequent section, we will introduce how we compute $\Delta\mathbf{P}_i$ with the aid of the boxes predicted at current layer.

**Canonical Object Spaces.** It is worth noting that the direction of the offsets are dependent on the definition of the world coordinate system and the object orientation, which complicates the learning of semantic position encoding. To address the limitation, we propose to transform it to an object coordinate system defined by the rotated bounding box. As illustrated in Figure 4, an offset vector in the world coordinate system can be transformed to the object coordinate system $(x_\theta, y_\theta)$ following:

$$\begin{bmatrix}\Delta x_\theta \\ \Delta y_\theta \\ \Delta z_\theta\end{bmatrix} = \begin{bmatrix}\cos\theta & -\sin\theta & 0 \\ \sin\theta & \cos\theta & 0 \\ 0 & 0 & 1\end{bmatrix}^T \begin{bmatrix}\Delta x \\ \Delta y \\ \Delta z\end{bmatrix} = \mathbf{R}_\theta^T \Delta\mathbf{p}, \quad (4)$$

where $\Delta\mathbf{p}$ is an element of $\Delta\mathbf{P}_i$. We use the other transformations in Equation 2 and Equation 3 to get the final normalized relative position bias item that models the rotated 3D bounding box position information. We perform 3DV-RPE operations for different Transformer decoder layers by default. We have observed that the canonical coordinate transformation bears resemblance to our earlier examination of Point-RCNN (Shi et al., 2019). In both instances, the bounding box undergoes a transformation into a canonical object space. However, a significant difference lies in the objectives of the canonical operation. In Point-RCNN, the objective is to achieve precise refinement by transforming the points contained within the 3D bounding box proposals. Conversely, our primary focus is on transforming the offset vectors to align with the object coordinate system.

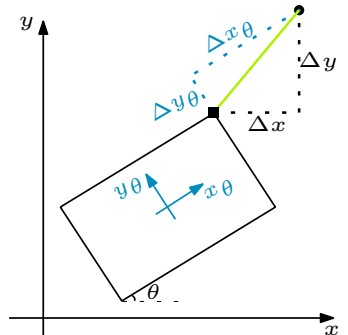

Figure 4: **Canonical object space transformation for 3DV-RPE.** The rectangle represents the box of the object, which defines an object coordinate system. The green line represents the offset from a point to the box vertex. The offset transformed to the object coordinate system is $(\Delta x_\theta, \Delta y_\theta)$ where the exact values can be geometrically reasoned. Since there is no rotation along the z-axis on the current datasets, we only show the changes in the x-y plane.

**Efficient implementation.** A naive implementation has high GPU memory consumption due to the large number of combinations between the object queries (each object query predicts a 3D bounding box) and the key-value points (output by the encoder), which makes it hard to train and deploy.

To solve this challenge, we use a smaller pre-defined 3DV-RPE table of shape: $\mathbf{T} \in \mathbb{R}^{10\times 10\times 10}$, which represents a discretized set of possible $\mathcal{F}(\Delta\mathbf{P}_i)$ that we interpolate into. Detailed information about 3DV-RPE table settings can be found in Appendix D. We do volumetric (5-D) grid_sample on the transformed 3DV-RPE table as follows:

$$\mathbf{P}_i = \mathrm{grid\_sample}(\mathrm{MLP}_i(\mathbf{T}), \ \mathcal{F}(\Delta\mathbf{P}_i)). \tag{5}$$

### 3.3 DETR WITH 3DV-RPE

**Framework.** We extend the original plain Transformer decoder, which consists of a stack of decoder layers and was designed for 2D object detection, to detect 3D bounding boxes from the irregular 3D points. Our approach has two steps: (i) as the first decoder layer has no access to coarse 3D bounding boxes, we employ a light-weight FFN to predict the initial 3D bounding boxes and feed

the top confident ones to the first Transformer decoder layer (e.g., $\{\theta^0, x^0, y^0, z^0, w^0, l^0, h^0\}$); and (ii) we update the bounding box predictions with the output of each Transformer decoder layer and use them to compute the modulation term in the multi-head cross-attention.

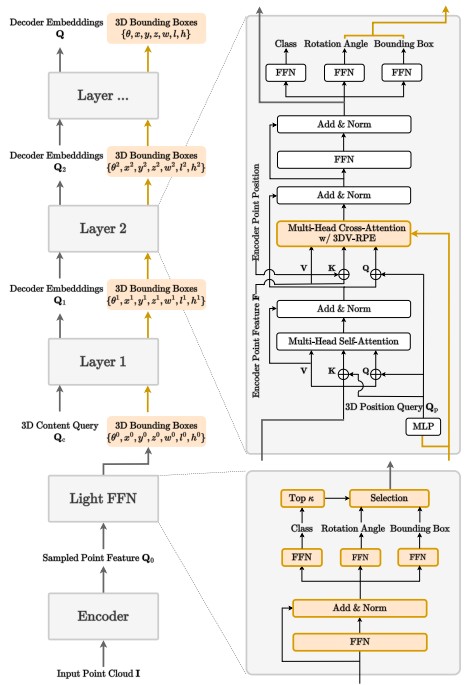

Figure 5: **Illustration of the proposed V-DETR framework.** We mark the modifications with yellow-colored regions and the other components, that are designed following the plain DETR, with gray-colored regions.

Figure 5 illustrates more details of the DETR with 3DV-RPE. For instance, we employ only the 3D content query $\mathbf{Q}_c$ as the input for the first decoder layer and use the decoder output embeddings $\mathbf{Q}^{i-1}$ from the $(i-1)$-th decoder layer as the input for the $i$-th decoder layer. We also apply MLP projects to compute the absolute position encodings of the 3D bounding boxes by default. We set the number of decoder layers as 8 following Misra et al. (2021). We predict the 3D bounding box delta target based on the initial prediction such as $\{\theta^0, x^0, y^0, z^0, w^0, l^0, h^0\}$ in all the Transformer decoder layers.

**Visualization.** Figure 6 shows the relative position attention maps learned with the 3DV-RPE scheme. We show the attention maps for 8 vertices in the first 4 columns and the merged ones in the last column. The visualization results show that (i) our 3DV-RPE can enhance the inner 3D box regions relative to each vertex position and (ii) combining the eight relative position attention maps can accurately localize the regions within the bounding box. We also show that 3DV-RPE can localize the extremity positions on the 3D object surface in the experiments.

## 4 EXPERIMENT

### 4.1 DATASETS AND METRICS

**Datasets.** We evaluate our approach on two challenging 3D indoor object detection benchmarks including:

*ScanNetV2* (Dai et al., 2017): ScanNetV2 consists of 3D meshes recovered from RGB-D videos captured in various indoor scenes. It has about 12K training meshes and 312 validation meshes, each annotated with semantic and instance segmentation masks for around 18 classes of objects. We follow Qi et al. (2019) to extract the point clouds from the meshes.

*SUN RGB-D* (Song et al., 2015): SUN RGB-D is a single-view RGB-D image dataset. It has about 5K images for both training and validation sets. Each image is annotated with oriented 3D bounding boxes for 37 classes of objects. We follow VoteNet (Qi et al., 2019) to convert the RGB-D image to the point clouds using the camera parameters and evaluate our approach on the 10 most common classes of objects.

**Metrics.** We report the standard mean Average Precision (mAP) under different IoU thresholds, $AP_{25}$ for 0.25 IoU threshold and $AP_{50}$ for 0.5 IoU threshold.

### 4.2 IMPLEMENTATION DETAILS

**Training.** We use the AdamW optimizer (Loshchilov & Hutter, 2019) with the base learning rate 7e-4, the batch size 8, and the weight decay 0.1. The learning rate is warmed up for 9 epochs, then is dropped to 1e-6 using the cosine schedule during the entire training process. We use gradient clipping to stabilize the training. We train for 360 epochs on ScanNetV2 and 240 epochs on SUN RGB-D in all experiments except for the system-level comparisons, where we train for 540 epochs on ScanNetV2. We use the standard data augmentations including random cropping (at least 30K points), random sampling (100K points), random flipping (p=0.5), random rotation along the z-axis

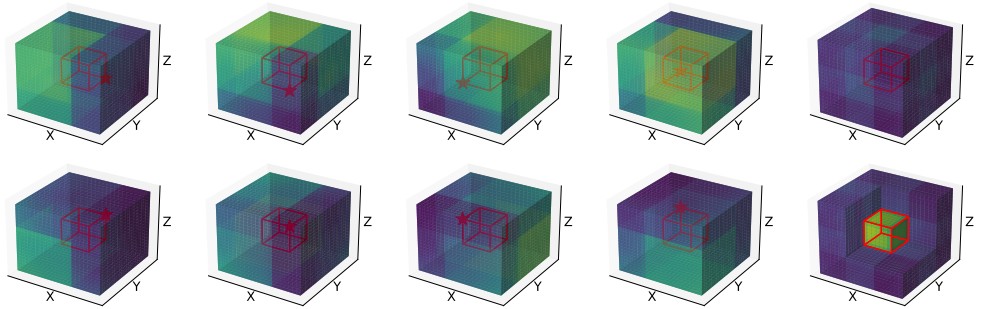

Figure 6: **Visualizing the learned spatial attention maps based on 3DV-RPE.** We use the small red-colored cube to represent the 3D bounding box of an object, the red five-pointed star to mark the eight vertices, and the entire colored cube as the input scene for simplicity. We average each $\mathbf{P}_i$ along head dimension according to Equation 2 and visualize eight vertices' learned spatial cross-attention maps (from column#1 to column#4). We visualize the merged spatial attention maps in column#5 (from the cutaway view). The color indicates the attention values: yellow for high and blue for low. We can observe that (i) the learned spatial attention maps of each vertex can enhance the regions along the internal direction starting from each vertex position, and (ii) the combined spatial attention maps can accurately enhance the internal regions inside the red-colored cubes.

$(-5°, 5°)$, random translation $(-0.4, 0.4)$, random scaling $(0.6, 1.4)$. We also use the one-to-many matching (Jia et al., 2022) to speed up the convergence speed with more rich and informative positive samples.

**Inference.** We process the entire point clouds of each scene and generate the bounding box proposals. We use 3D NMS to suppress the duplicated proposals in the one-to-many matching setting, which is not needed in the one-to-one matching setting. We also use test-time augmentation, i.e., flipping, by default unless specified otherwise.

## 4.3 ANALYSIS OF PERFORMANCE ENHANCEMENTS BY 3DV-RPE

Table 1: Effect of the position encoding method choices. Numbers in ( ) represent results without TTA

| position encoding method | #epochs | $AP_{25}$ | $AP_{50}$ |
|---|---|---|---|
| None (Our Baseline) | 360 | 68.8(67.9) | 44.5(43.5) |
| 3D box mask | 360 | 74.0(72.9) | 59.1(58.3) |
| 3DV-RPE | 360 | 76.7(76.2) | 65.0(64.2) |
| 3D box mask + 3DV-RPE | 360 | 76.0(75.3) | 62.7(61.5) |
| None (Our Baseline) | 540 | 71.4(70.6) | 47.6(46.7) |
| 3D box mask | 540 | 75.1(74.2) | 60.8(59.6) |
| 3DV-RPE | 540 | 77.8(77.4) | 66.0(65.0) |
| 3D box mask + 3DV-RPE | 540 | 77.0(76.5) | 63.5(62.4) |

Table 2: Implementation of 3DV-RPE on 3DETR.

| method | #epochs | $AP_{25}$ | $AP_{50}$ |
|---|---|---|---|
| 3DETR | 1080 | 65.0 | 47.0 |
| 3DETR + 3DV-PRE | 1080 | 71.2 | 60.8 |

Table 3: Implementation of 3DV-RPE on Group-Free.

| method | #epochs | $AP_{25}$ | $AP_{50}$ |
|---|---|---|---|
| Group-Free | 400 | 69.1 | 52.8 |
| Group-Free + 3DV-PRE | 400 | 72.8 | 62.1 |

**Different position encoding methods.** Table 1 compares our 3DV-RPE on ScanNetV2 with another position encoding method – 3D box mask, which sets the relative position encoding term to $-\infty$ for positions outside the 3D bounding box and $0$ otherwise. The first and fourth rows essentially display the results of our baseline settings, which have been augmented with various additional enhancements introduced in the section 3.1. The results show that (i) the 3D box mask method achieves strong results on $AP_{25}$, and (ii) our 3DV-RPE significantly improves over the 3D box mask method on $AP_{50}$. We speculate that our 3DV-RPE performs better because the 3D box mask method suffers from error accumulation from the previous decoder layers and cannot be optimized end-to-end. We also report the results of combining the 3D box mask and 3DV-RPE, which performs better than the 3D box mask scheme but worse than our 3DV-RPE. This verifies that our 3DV-RPE can learn to (i) exploit more accurate geometric structure information within the 3D bounding box and (ii) benefit from capturing useful long-range context information outside the box. Moreover, we report the results with longer training epochs and observe that the gap between the 3D box mask and 3DV-RPE remains, thus further demonstrating the advantages of our approach.

Table 4: System-level comparison with the state-of-the-art on ScanNetV2 and SUN RGB-D. TTA: test-time augmentation. * denotes using additional self-supervised pre-training. ** denotes using extra training data.

| Method | ScanNetV2 | | SUN RGB-D | |
|---|---|---|---|---|
| | $AP_{25}$ | $AP_{50}$ | $AP_{25}$ | $AP_{50}$ |
| 3DETR (Misra et al., 2021) | 65.0 | 47.0 | 59.1 | 32.7 |
| Group-Free (Liu et al., 2021b) | 69.1 | 52.8 | 63.0 | 45.2 |
| AShapeFormer (Li et al., 2023) | 71.1 | 56.6 | 62.2 | – |
| FCAF3D (Rukhovich et al., 2022) | 71.5 | 57.3 | 64.2 | 48.9 |
| OctFormer (Wang, 2023) | – | – | 66.2 | 50.6 |
| Uni3DETR (Wang et al., 2023) | 71.7 | 58.3 | 67.0 | 50.3 |
| ConDaFormer (Duan et al., 2023) | – | – | 67.1 | 49.9 |
| TR3D (Rukhovich et al., 2023) | 72.9 | 59.3 | 67.1 | 50.4 |
| Point-GCC* (Fan et al., 2023a) | 73.1 | 59.6 | 67.7 | 51.0 |
| CAGroup3D (Wang et al., 2022a) | 75.1 | 61.3 | 66.8 | 50.2 |
| SWIN3D + CAGroup3D** (Yang et al., 2023) | 76.4 | 63.2 | – | – |
| V-DETR | 77.4 | 65.0 | 67.5 | 50.4 |
| V-DETR (TTA) | **77.8** | **66.0** | **68.0** | **51.1** |
| *Average Results under 25× trials* | | | | |
| Group-Free (Liu et al., 2021b) | 68.6 | 51.8 | 62.6 | 44.4 |
| FCAF3D (Rukhovich et al., 2022) | 70.7 | 56.0 | 63.8 | 48.2 |
| TR3D (Rukhovich et al., 2023) | 72.0 | 57.4 | 66.3 | 49.6 |
| CAGroup3D (Wang et al., 2022a) | 74.5 | 60.3 | 66.4 | 49.5 |
| ConDaFormer (Duan et al., 2023) | – | – | 66.8 | 49.5 |
| V-DETR | 76.8 | 64.5 | 66.8 | 49.7 |
| V-DETR (TTA) | **77.0** | **65.3** | **67.5** | **50.0** |

**Implementation of 3DV-RPE on other baselines.** In order to verify the universality and usefulness of 3DV-RPE, we implement our method on other baselines, such as 3DETR (Misra et al., 2021) and Group-Free (Liu et al., 2021b).The comparative analysis, as illustrated in Table 2 and Table 3, demonstrates the performance impact of integrating 3DV-RPE on the ScanNetV2 benchmark under equitable conditions. It is evident from the results that the incorporation of our 3DV-RPE significantly improves performance, notably in $AP_{50}$, in different baselines.

*More analyses.* We offer additional comparative analyses between our 3DV-RPE with absolute position encoding methods and other advanced attention modulation methods, to substantiate the superiority of our approach.The detailed results of these comparisons are accessible in Appendix C.

## 4.4 COMPARISONS WITH PREVIOUS SYSTEMS

In Table 4, we compare our method with the state-of-the-art methods of most recent works at the system level, and a more detailed comparison with more previous work can be found in Appendix B. These methods use different techniques, so we cannot compare them in a controlled way. According to the results, we show that our method performs the best, either measured by the highest performance or the average results under multiple trials. For example, on ScanNetV2 val set, our method achieves $AP_{25}$=77.8% and $AP_{50}$=66.0%, which surpasses the latest state-of-the-art SWIN3D + CAGroup3D (Yang et al., 2023) that reports $AP_{25}$=76.4% and $AP_{50}$=63.2% while SWIN3D requires additional data for pertaining. Notably, on ScanNetV2, we observe more significant gains on $AP_{50}$ (+2.8%) that requires more accurate localization, i.e., under a higher IoU threshold. We also observe consistent gains on both $AP_{25}$ and $AP_{50}$ on SUN RGB-D.

## 4.5 3DV-RPE ABLATION EXPERIMENTS

We conduct all the following ablation experiments on ScanNetV2 and trained with 360 epochs for efficiency, except for the ablation experiments on the coordinate system, where we report the results on SUN RGB-D.

**Number of vertex.** Table 5 shows the effect of different numbers of vertices for computing the relative position bias term. The results show that using 8 vertices performs the best, so we use this setting by default. We attribute their close performances to the fact that they essentially share the same minimal and maximal XYZ values when using fewer vertices such as 2 or 4, which is caused by the zero rotation angles on ScanNetV2.

Table 5: Effect of the number of vertex within 3DV-RPE.

| # vertex | $AP_{25}$ | $AP_{50}$ |
|---|---|---|
| 1 | 73.4 | 54.8 |
| 2 | 76.1 | 63.1 |
| 4 | 76.3 | 63.4 |
| 8 | 76.7 | 65.0 |

Table 6: Effect of non-linear transform within 3DV-RPE.

| $\mathcal{F}(\cdot)$ | $AP_{25}$ | $AP_{50}$ |
|---|---|---|
| $\mathcal{F}(x) = x$ | 69.6 | 48.2 |
| $\mathcal{F}(x) = x/(1 + \|x\|)$ | 76.0 | 62.6 |
| $\mathcal{F}(x) = \tanh(x)$ | 76.3 | 62.6 |
| $\mathcal{F}(x) = x/\sqrt{1 + x^2}$ | 76.6 | 63.0 |
| $\mathcal{F}(x) = \text{sign}(x)\log(1 + \|x\|)$ | 76.7 | 65.0 |

Table 7: Effect of the coordinate system on SUN RGB-D.

| coordinate system | $AP_{25}$ | $AP_{50}$ |
|---|---|---|
| world coord. | 65.8 | 46.9 |
| object coord. | 68.0 | 51.1 |

Table 8: Effect of the object-normalized box parameterization.

| object-normalize | $AP_{25}$ | $AP_{50}$ |
|---|---|---|
| ✗ | 74.9 | 61.1 |
| ✓ | 76.7 | 65.0 |

**Non-linear transform.** Table 6 shows the effect of different non-linear transform functions. The results show that the signed log function performs the best. The signed log function magnifies small changes in smaller ranges. Therefore, we choose the signed log function by default.

**Coordinate system on SUN RGB-D.** We evaluate the effect of the coordinate system on calculating the relative positions in our 3DV-RPE on SUN RGB-D, which requires predicting the rotation angle along the $z$-axis. Table 7 shows the results. We find that transforming the relative offsets from the world coordinate system to the object coordinate system significantly improves the performance, e.g., $AP_{25}$ and $AP_{50}$ increase by +2.2% and 4.2%, respectively.

**Object-normalized box parameterization.** In Table 8, we show the effect of using object-normalized box parameterization. We find using the object-normalized scheme significantly boosts the $AP_{50}$ from 61.1 to 65.0.

**Qualitative comparisons.** We show some examples of V-DETR detection results on SUN RGB-D in Figure 7 and on ScanNetV2 in Figure 8 in the Appendix E, where the scenes are diverse and challenging with clutter, partiality, scanning artifacts, etc. Our V-DETR performs well despite these challenges.

**Encoder choice.** Table 9 compares the results of using different encoder architectures. We find that using a sparse 3D version of ResNet34 with an FPN neck achieves the best results. Therefore, we use ResNet34 + FPN as our default encoder.

Table 9: Effect of the encoder choice.

| encoder | $AP_{25}$ | $AP_{50}$ |
|---|---|---|
| PointNet + Tran.Enc. | 73.6 | 60.1 |
| ResNet34 + FPN | 76.7 | 65.0 |

*More ablation experiments.* We provide more ablation studies on the effects of using different shapes for the pre-defined 3DV-RPE table, one-to-many matching, the number of points, and other factors in the Appendix A.

## 5 CONCLUSION

In this work, we have shown how to make DETR-based approaches competitive for indoor 3D object detection tasks. The key contribution is an effective 3D vertex relative position encoding (3DV-RPE) scheme that can model the accurate position information in the irregular sparse 3D point cloud directly. We demonstrate the advantages of our approach by achieving strong results on two challenging 3D detection benchmarks. We also plan to extend our approach to outdoor 3D object detection tasks, which differ from most existing methods that rely on modern 2D DETR-based detectors by converting 3D points to a 2D bird-eye-view plane. We hope our approach can show the potential for unifying the object detection architecture design for indoor and outdoor 3D detection tasks.

## ACKNOWLEDGEMENT

We thank all the anonymous reviewers and Area Chairs for their constructive and helpful comments, which have significantly improved the quality of the paper. The work was partly supported by the National Natural Science Foundation of China (Grant No. 62088102).

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

# A  MORE ABLATION EXPERIMENTS AND ANALYSIS

Table 10: Effect of light-weight FFN.

| Light-weight FFN | $AP_{25}$ | $AP_{50}$ |
|:---:|:---:|:---:|
| ✗ | 76.6 | 62.8 |
| ✓ | 76.7 | 65.0 |

Table 11: Effect of using more points.

| # of points | $AP_{25}$ | $AP_{50}$ |
|:---:|:---:|:---:|
| 20K | 73.9 | 61.6 |
| 40K | 75.2 | 62.4 |
| 100K | 76.7 | 65.0 |

Table 12: Effect of the one-to-many matching.

| # points | #query | #repeat number | $AP_{25}$ | $AP_{50}$ |
|:---:|:---:|:---:|:---:|:---:|
| | 256 | 1 | 74.3 | 62.0 |
| 40K | 512 | 2 | 75.6 | 62.9 |
| | 1024 | 4 | 75.2 | 62.4 |
| | 256 | 1 | 75.3 | 63.7 |
| 100K | 512 | 2 | 76.4 | 64.2 |
| | 1024 | 4 | 76.7 | 65.0 |

Table 13: Effect of the pre-defined 3DV-RPE table shape.

| 3DV-RPE table shape | $AP_{25}$ | $AP_{50}$ |
|:---:|:---:|:---:|
| $5 \times 5 \times 5$ | 76.7 | 64.7 |
| $10 \times 10 \times 10$ | 76.7 | 65.0 |
| $25 \times 25 \times 25$ | 76.7 | 64.2 |
| $50 \times 50 \times 50$ | 76.7 | 64.3 |

Table 14: Effect of voxel expansion.

| method | voxel expansion | $AP_{25}$ | $AP_{50}$ |
|:---:|:---:|:---:|:---:|
| FCAF3D | ✗ | 67.5 | 52.4 |
| | ✓ | 70.5 | 54.8 |
| Ours | ✗ | 76.7 | 65.0 |
| | ✓ | 75.5 | 62.0 |

Table 15: Inference cost comparison. We evaluate all numbers on a Tesla V100 PCIe 16 GB GPU with batch size as 1 for a fair comparison.

| method | # Scenes/second | Latency/scene | GPU Memory |
|:---|:---:|:---:|:---:|
| FCAF3D | 7.8 | 128ms | 628M |
| CAGroup3D | 2.1 | 480ms | 1138M |
| Ours (light) | 7.7 | 130ms | 489M |
| Ours | 4.2 | 240ms | 642M |

**Light-weight FFN.** Table 10 reports the comparison results on the effect of proposed light FFN. According to the results, we observe that using the light-weight FFN significantly boosts the $AP_{50}$ from 62.8 to 65.0, thus showing the advantages of using a set of adaptive predicted initial 3D bounding boxes over a set of pre-defined 3D bounding boxes of the same size.

**Number of points during training and evaluation.** In Table 11, we report the comparison results when using different number of points during training. We observe that using 100K points achieves consistently better performance, thus we choose 100K points.

**One-to-many matching.** Table 12 shows the comparison results when choosing different hyper-parameters for a one-to-many matching scheme. For example, we find increasing the number of queries and the number of ground truth repeating times even hurts the performance when training with 40K points but improves the performance when training with 100K.

**Table shape.** In Table 13, we show the effect of different shapes for the pre-defined 3DV-RPE table. We find that $10 \times 10 \times 10$ achieves the best results. Our approach is less sensitive to the shape of the 3DV-RPE table thanks to the signed log function, which improves the interpolation quality to some degree.

**Voxel expansion.** Table 14 evaluates the effect of using voxel expansion in the FPN neck when the encoder is ResNet34 + FPN. We also compare our results with the recent FCAF3D method. The results show that (i) voxel expansion is crucial for FCAF3D, which relies on building virtual center features; and (ii) voxel expansion degrades the performance when using DETR, which might lose the original accurate 3D surface information. Therefore, we demonstrate an important advantage of using DETR-based approaches, i.e., they do not require complicated voxel expansion operations.

**Inference complexity comparison.** Table 15 reports the comparison results to FCAF3D and CA-Group3D. We do not apply the test-time augmentation (TTA) to ensure fair comparisons. Accordingly, our method achieves a better performance-efficiency trade-off than CAGroup3D. We also provide a light version by decreasing the number of 3D object query from 1024 to 256, which achieves

$AP_{25}$=75.6 and $AP_{50}$=62.7. Notably, the reported latency of CAGroup3D is close to the numbers in their official logs but different from the numbers reported in the paper (179.3ms tested on RTX 3090 GPU). The authors of CAGroup3D have acknowledged this issue in their GitHub repository.

# B MORE DETAILED SYSTEM-LEVEL COMPARISON

Table 16: Full System-level comparison with the state-of-the-art on ScanNetV2 and SUN RGB-D. TTA: test-time augmentation. * denotes using additional self-supervised pre-training. ** denotes using extra training data.

| Method | ScanNetV2 | | SUN RGB-D | |
|---|---|---|---|---|
| | $AP_{25}$ | $AP_{50}$ | $AP_{25}$ | $AP_{50}$ |
| VoteNet (Qi et al., 2019) | 58.6 | 33.5 | 57.7 | - |
| HGNet (Chen et al., 2020) | 61.3 | 34.4 | 61.6 | - |
| 3D-MPA (Engelmann et al., 2020) | 64.2 | 49.2 | - | - |
| MLCVNet (Xie et al., 2020) | 64.5 | 41.4 | 59.8 | - |
| GSDN (Gwak et al., 2020) | 62.8 | 34.8 | - | - |
| H3DNet (Zhang et al., 2020) | 67.2 | 48.1 | 60.1 | 39.0 |
| BRNet (Cheng et al., 2021) | 66.1 | 50.9 | 61.1 | 43.7 |
| 3DETR (Misra et al., 2021) | 65.0 | 47.0 | 59.1 | 32.7 |
| VENet (Xie et al., 2021) | 67.7 | - | 62.5 | 39.2 |
| Group-Free (Liu et al., 2021b) | 69.1 | 52.8 | 63.0 | 45.2 |
| RBGNet (Wang et al., 2022b) | 70.6 | 55.2 | 64.1 | 47.2 |
| HyperDet3D (Zheng et al., 2022) | 70.9 | 57.2 | 63.5 | 47.3 |
| AShapeFormer (Li et al., 2023) | 71.1 | 56.6 | 62.2 | − |
| FCAF3D (Rukhovich et al., 2022) | 71.5 | 57.3 | 64.2 | 48.9 |
| OctFormer (Wang, 2023) | − | − | 66.2 | 50.6 |
| Uni3DETR (Wang et al., 2023) | 71.7 | 58.3 | 67.0 | 50.3 |
| ConDaFormer (Duan et al., 2023) | − | − | 67.1 | 49.9 |
| TR3D (Rukhovich et al., 2023) | 72.9 | 59.3 | 67.1 | 50.4 |
| Point-GCC* (Fan et al., 2023a) | 73.1 | 59.6 | 67.7 | 51.0 |
| CAGroup3D (Wang et al., 2022a) | 75.1 | 61.3 | 66.8 | 50.2 |
| SWIN3D + CAGroup3D** (Yang et al., 2023) | 76.4 | 63.2 | − | − |
| V-DETR | 77.4 | 65.0 | 67.5 | 50.4 |
| V-DETR (TTA) | **77.8** | **66.0** | **68.0** | **51.1** |
| *Average Results under $25\times$ trials* | | | | |
| Group-Free (Liu et al., 2021b) | 68.6 | 51.8 | 62.6 | 44.4 |
| RBGNet (Wang et al., 2022b) | 69.9 | 54.7 | 63.6 | 46.3 |
| FCAF3D (Rukhovich et al., 2022) | 70.7 | 56.0 | 63.8 | 48.2 |
| TR3D (Rukhovich et al., 2023) | 72.0 | 57.4 | 66.3 | 49.6 |
| CAGroup3D (Wang et al., 2022a) | 74.5 | 60.3 | 66.4 | 49.5 |
| ConDaFormer (Duan et al., 2023) | − | − | 66.8 | 49.5 |
| V-DETR | 76.8 | 64.5 | 66.8 | 49.7 |
| V-DETR (TTA) | **77.0** | **65.3** | **67.5** | **50.0** |

Table 16 shows a more detailed comparison of our method with the state-of-the-art methods from previous works at the system level.

# C MORE ANALYSIS OF 3DV-RPE ADVANTAGES

**Comparison with other attention modulation methods.** We summarize the comparison results with other advanced related methods including contextual relative position encoding (CRPE) (Lai et al., 2022; Yang et al., 2022), conditional cross-attention (Cond-CA) (Meng et al., 2021), dynamic anchor box cross-attention (DAB-CA) (Liu et al., 2022) in Table 17. We report the comparison results under the most strong settings, i.e., 540 training epochs. Accordingly, we see that (i) both CRPE (Stratified Transformer (Lai et al., 2022)) and CRPE (EQNet (Yang et al., 2022)) consistently

improve the baseline; (ii) our 3DV-RPE achieves the best performance. The reason is that the CRPE methods of Stratified-Transformer (Lai et al., 2022) and EQNet (Yang et al., 2022) only consider the center point of the 3D box, while our 3DV-RPE explicitly considers the $8\times$ vertex points and rotated angle of the 3D box. Our method encodes the box size and the six faces, thus modeling the accurate position relations between all other points and the 3D bounding box (supported by the much larger gains on $AP_{50}$).

Table 17: Comparison to other attention modulation methods. We only change the decoder cross-attention scheme and keep all other settings the same for comparison fairness.

| method | #epochs | $AP_{25}$ | $AP_{50}$ |
|---|---|---|---|
| Baseline (w/o RPE) | 540 | 71.4 | 47.6 |
| Baseline + CRPE (Stratified Transformer) | 540 | 74.7 | 58.1 |
| Baseline + CRPE (EQNet) | 540 | 73.1 | 54.4 |
| Baseline + Cond-CA | 540 | 74.7 | 55.8 |
| Baseline + DAB-CA | 540 | 75.4 | 56.0 |
| Baseline + 3DV-RPE | 540 | 77.8 | 66.0 |

**Comparison with absolute position encoding methods on ScanNetV2.** We summarize the comparison results with different absolute position encoding(APE) methods, including the sin-cos absolute position encoding (APE w/ Sin-Cos) and the absolute position encoding using a multilayer perceptron (APE w/ MLP). To minimize the gap between APE and 3DV-RPE for fair compression, we apply non-linear transformation and predefined table to APE, mirroring their usage in 3DV-RPE. As depicted in Table 18, our 3DV-RPE significantly outperforms all the APE variants by a large margin, especially on AP50 which demands greater precision in localization.

Table 18: Comparison to absolute position encoding(APE) methods on ScanNetV2. "NonLinear" refers to the application of non-linear transformation, and "Predefined Table" denotes employing table and grid sampling techniques to enhance efficiency, which are both applied to 3DV-RPE.

| method | #epochs | $AP_{25}$ | $AP_{50}$ |
|---|---|---|---|
| Baseline (w/o PE) | 540 | 71.4 | 47.6 |
| Baseline + APE w/ Sin-Cos | 540 | 71.8 | 47.9 |
| Baseline + APE w/ MLP + NonLinear | 540 | 72.1 | 48.7 |
| Baseline + APE w/ MLP + NonLinear + Predefined Table | 540 | 72.0 | 48.5 |
| Baseline + 3DV-RPE | 540 | 77.8 | 66.0 |

## D  DETAILS ABOUT 3DV-RPE EFFICIENT IMPLEMENTATION

The transformation matrix $\mathbf{T}$ plays a pivotal role in our proposed method, facilitating the efficient encoding of spatial relations. The matrix is initialized to encompass a $N_{\text{table}} \times N_{\text{table}} \times N_{\text{table}}$ grid, i.e., $N_{\text{table}} = 10$, where each grid point $(i, j, k)$ holds a three-dimensional vector. The vector values are calculated as follows:

$$\mathbf{T}[i,j,k] = \left( \frac{M(2i - (N_{\text{table}} - 1))}{N_{\text{table}} - 1}, \frac{M(2j - (N_{\text{table}} - 1))}{N_{\text{table}} - 1}, \frac{M(2k - (N_{\text{table}} - 1))}{N_{\text{table}} - 1} \right), \quad (6)$$

where $[-M, M]$ is the range of $\mathbf{T}$ values, which avoids $\mathcal{F}(\Delta \mathbf{P}_i)$ in Equation 5 beyond the bounds.

# E  QUALITATIVE RESULTS AND ANALYSIS

We show some qualitative examples of our V-DETR detection on SUN RGB-D and ScanNetV2 in Figure 7 and Figure 8, respectively. We can observe that our method can find most of the target objects in various scenes.

Figure 9 shows the spatial cross-attention maps of our 3DV-RPE on three ScanNetV2 scenes. We see that (i) our 3DV-RPE can find the 3D bounding boxes accurately and (ii) each vertex's RPE can enhance the regions inside the boxes from that vertex.

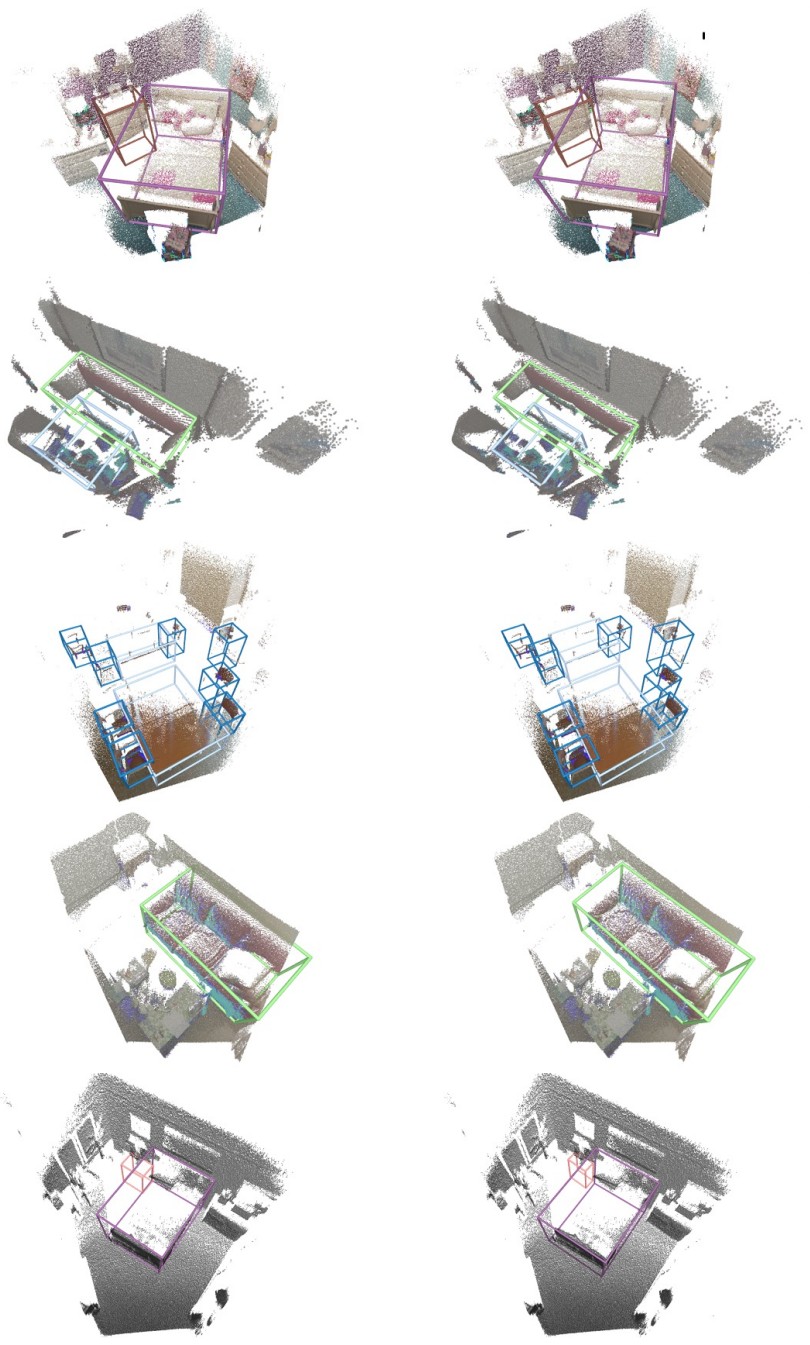

Figure 7:  More qualitative results of 3D object detection on SUN RGB-D. The ground truth is shown in the first column and our method's detection results are shown in the second column.

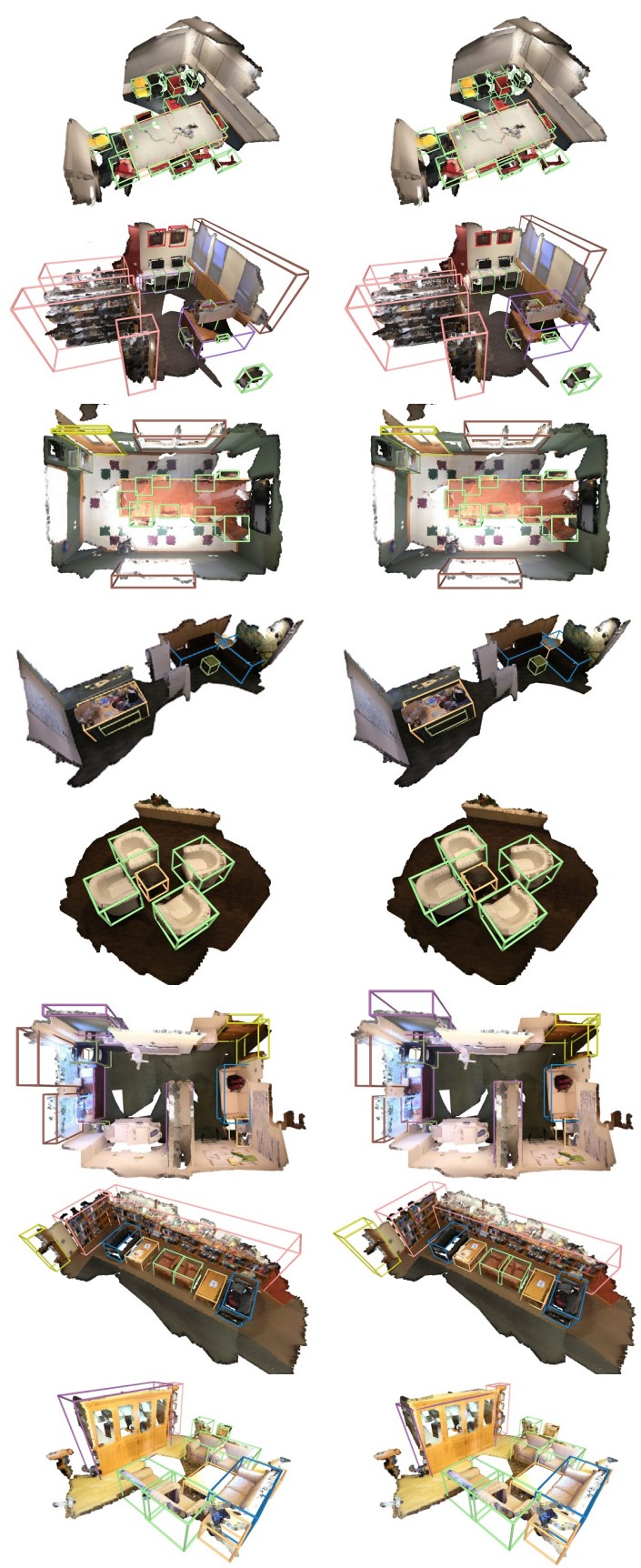

Figure 8: More qualitative results of 3D object detection on ScanNetV2. The ground-truth is shown in the first column and our method's detection results are shown in the second column.

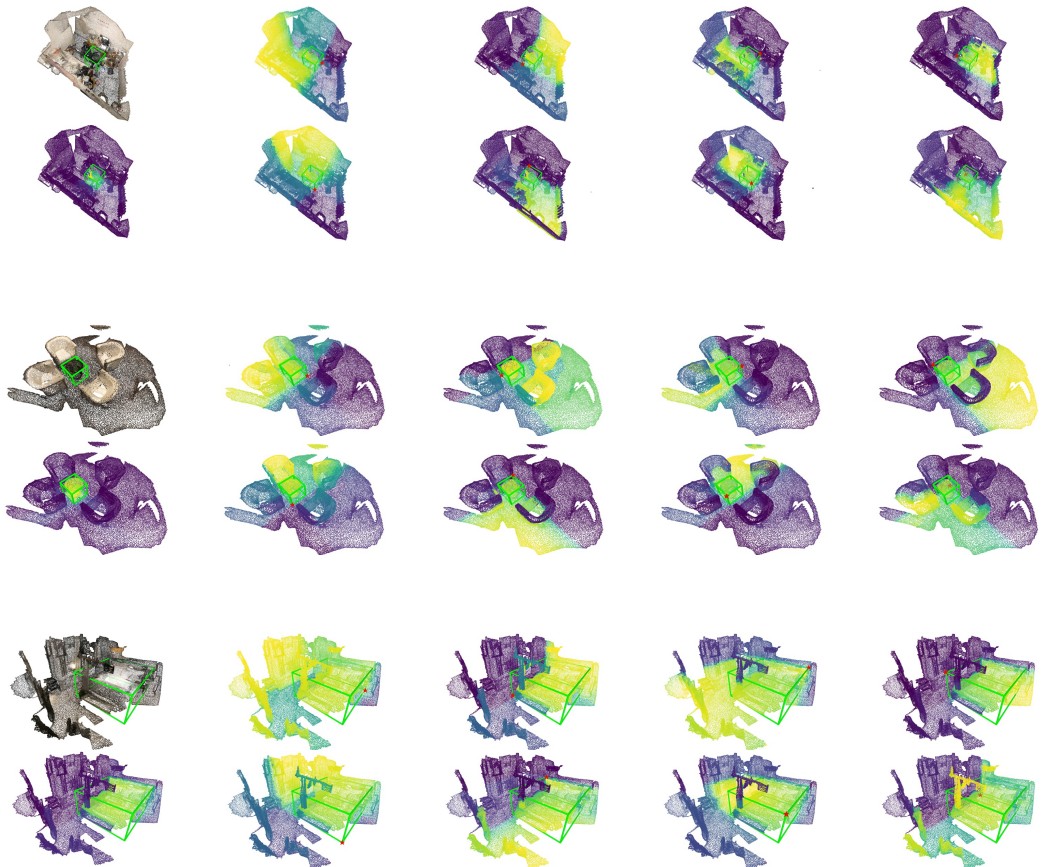

Figure 9: **Illustration of the spatial attention maps learned by our 3DV-RPE on ScanNetV2 scenes.** Each scene consists of two rows. We draw a green cube to mark the detected 3D bounding box and a red star at its eight vertices. We average the head dimension of each $\mathbf{P}_i$ and show the spatial cross-attention maps for eight vertices (columns 2-5). Column 1 shows the input scene and the merged attention maps. The color shows the attention values: yellow is high and blue is low. We see that (i) each vertex's attention map highlights the regions inside the cube from that vertex, and (ii) the combined attention maps focus on the regions inside the red cubes.

## F EFFECT OF DATA SCALE ON LEARNING THE LOCALITY INDUCTIVE BIAS FOR 2D OBJECT DETECTION

We conduct experiments on the 2D detector DETR Carion et al. (2020), using approximately $1\%$ of the training data ($1,200$ images), following your suggestion. We train DETR for the same number of iterations ($2,217,881$ iterations) as the original DETR, which was trained on the full dataset for $\sim 300$ epochs, while maintaining a batch size of 16. First, we observe that the mAP of the validation set drops from $44.9\%$ to $10.4\%$, which closely aligns with the performance reported in Table 4 of DETReg Bar et al. (2022).

Second, in accordance with Figure 6 of the DETR paper, we visualize the cross-attention maps for the predicted object in Figure 10. We note that these attention maps fail to focus on local object regions, especially the object extremities. Therefore, it is evident that the scale of data has a substantial impact on the model's ability to effectively learn the locality inductive biases.

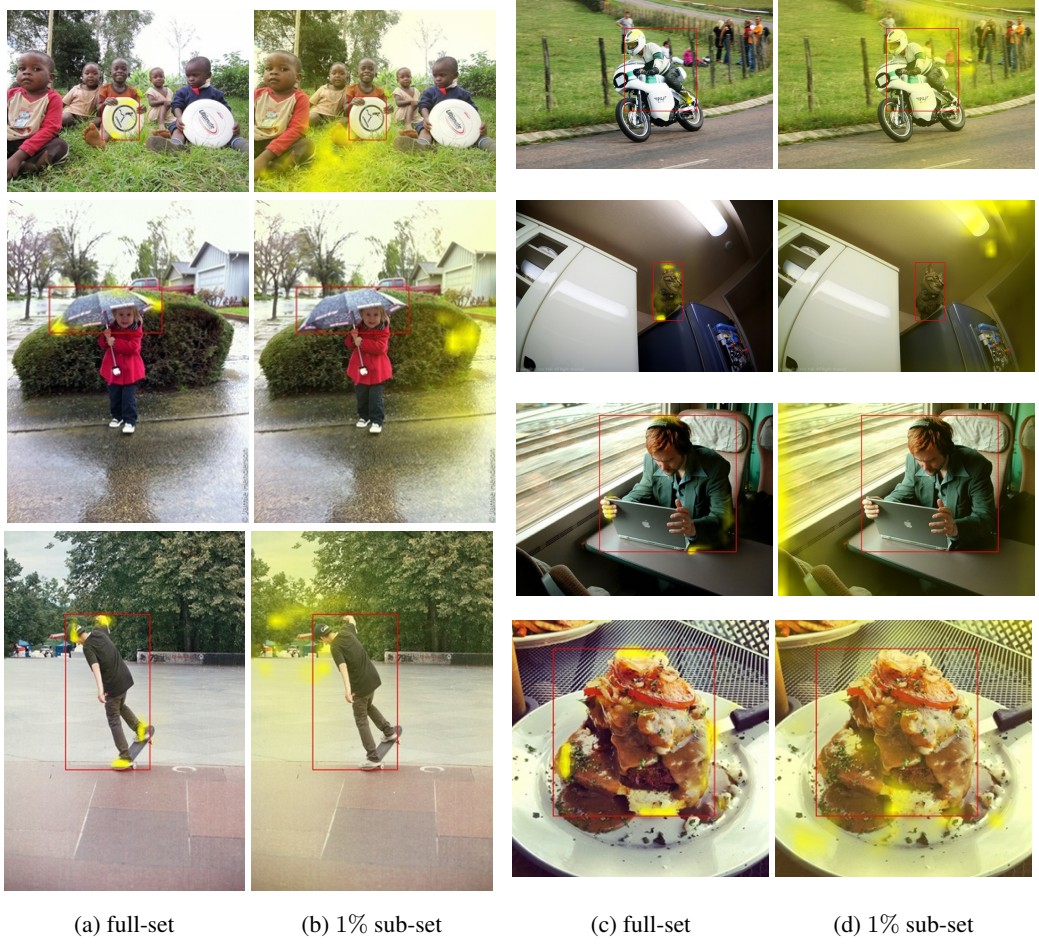


(a) full-set      (b) 1% sub-set      (c) full-set      (d) 1% sub-set


Figure 10: Illustrating the effect of training data scale on the 2D cross-attention maps with DETR. When trained using just a $1\%$ subset, it is observable that these attention maps struggle to concentrate on local object areas, particularly the object extremities.

