# OpenReview forum: "V-DETR: DETR with Vertex Relative Position Encoding for 3D Object Detection"
_ICLR.cc/2024/Conference — ICLR 2024 poster_

### Official Review · Reviewer_fSn3 · 2023-10-29

**Soundness:** 3 good
**Presentation:** 2 fair
**Contribution:** 3 good
**Rating:** 6
**Confidence:** 3

**Summary:**

This paper introduces an effective enhancement to DETR-based indoor 3D object detection. The key idea is to add a relative positional embedding between the queries and points. The positional embedding, which is called 3D Vertex Relative Position Encoding (3DV-RPE), calculates the relative positional embedding under the coordinate system of each 3D bounding box generated from the query. After incorporating the positional embedding, the performance significantly increases on both ScanNetv2 and Sun-RGBD.

**Strengths:**

The paper has demonstrated the following strengths:

* The approach of the paper has good support from their performance improvement on ScanNetv2 and Sun-RGBD.
* The vertex relative position encoding (3DV-RPE) has the reasonable intuition of embedding position information for 3D detection and will likely inspire other readers.
* The qualitative results like Figure 1 clearly illustrate the implication of the method in the paper in guiding models' attention.

**Weaknesses:**

* I suggest improving the order of presentation in Sec. 3.2 and Sec. 3.2. For example, I suggest moving the paragraph of "3DV-RPE" before talking about "canonical object space" and other details. When I read this part, I was quite confused by Sec. 3.2, not knowing how $R$ is generated, what is $P_i$, etc.

* As position encoding is the focus of this paper, I expect the authors to analyze or conduct ablation studies on more position encoding algorithms. Details are in the "questions" section below.

* I also haven't found the performance of the baseline without relative position encoding. In case I missed it, I suggest the authors put it into Sec. 4.3 or Sec. 4.4 for a clear ablation study.

Typo on page 5, line 1: fig:rotatedRPE

**Questions:**

1. **Baseline performance.** As mentioned in the weakness section, could you remind me where you have put the baseline performance? It is critical to recognize the improvement of 3DV-RPE. Technically, I wish to see that under the same normalization and other tricks, 3DV-RPE is indeed helpful.


2. **Additional analysis.** With position encoding being the center of this paper, I think it necessary to conduct ablation studies on other common formats of position encoding, such as:
*  Absolute position encoding, in both the formats you proposed like Eqn. 3 and Eqn. 4, or common sin-cos position encoding.
* More justifications of hyper-parameters. For example, where does 10 come from in $T$'s shape? May I use another number to replace 10?

---

> ### Author Response · Authors · 2023-11-21
> **Response to Reviewer fSn3 (1/2)**
>
> We thank the reviewer for the careful reviews and constructive suggestions. We answer the questions as follows.
>
> > **Q1. I suggest improving the order of presentation in Sec. 3.2 and Sec. 3.2. For example, I suggest moving the paragraph of "3DV-RPE" before talking about "canonical object space" and other details. When I read this part, I was quite confused by Sec. 3.2, not knowing how $\mathbf{R}$ is generated, what is $\mathbf{P}_{i}$, etc.**
>
> Great suggestion! Thanks for pointing out this issue. **We have revised the organization of Sec. 3.1 and Sec. 3.2 to make it easier to follow. Please check the blue text in the revised PDF.**
>
> > **Q2. As position encoding is the focus of this paper, I expect the authors to analyze or conduct ablation studies on more position encoding algorithms. Details are in the "questions" section below.**
>
> Great suggestions! First, we would like to invite you to refer to Table 13 and the related discussion in our submission. We guess the comparison results with the latest position encoding algorithms (in the first two rows) might be the mentioned ablation studies on more position encoding algorithms. Besides, we also answer your other detailed concerns in the following response.
>
> For your convenience, we also attached Table 13 in the following:
>
> | method                                   | epoch | AP25 | AP50 |
> | ---------------------------------------- | ----- | ---- | ---- |
> | Baseline (w/o RPE)                       | 540x   | 71.4 | 47.6 |
> | Baseline + CRPE (Stratified Transformer[1]) | 540x   | 74.7 | 58.1 |
> | Baseline + CRPE (EQNet[2])                  | 540x   | 73.1 | 54.4 |
> | Baseline + 3DV-RPE                       | 540x   | 77.8 | 66.0 |
>
> Accordingly, our 3DV-RPE significantly outperforms these two advanced RPE schemes, especially on AP50 metrics (with gains of +7.9 and +11.6 respectively). We observe that both the CRPE (Stratified Transformer) and CRPE (EQNet) can achieve performance gains similar to a special case of our 3DV-RPE that only considers the 3D bounding box center (AP25=73.4/AP50=54.8). Therefore, it's clear that the key to effectiveness is to explicitly consider the relative position information between each point and the 8 vertex points of a given 3D bounding box. We will add the above information in the final revision.
>
> [1] Stratified Transformer for 3D Point Cloud Segmentation, CVPR 2022
>
> [2] A Unified Query-based Paradigm for Point Cloud Understanding, CVPR 2022
>
> > **Q3. I also haven't found the performance of the baseline without relative position encoding. In case I missed it, I suggest the authors put it into Sec. 4.3 or Sec. 4.4 for a clear ablation study.**
>
> Good point! In fact, we have reported the performance of the baseline without relative position encoding in Table 6 (1st row and 5th row). We would like to reorganize Sec. 4.4 to place Table 6 and the related discussion at the forefront to make it clearer.
>
> We also provide the detailed comparison results on ScanNet for reference:
>
> |                        | \#epoch | AP25 | AP50 |
> | ---------------------- | ------- | ---- | ---- |
> | Our Baseline           | 540x    | 71.4 | 47.6 |
> | Our Baseline + 3D-VPRE | 540x    | 77.8 | 66.0 |
>
> > **Q4. Baseline performance. As mentioned in the weakness section, could you remind me where you have put the baseline performance? It is critical to recognize the improvement of 3DV-RPE. Technically, I wish to see that under the same normalization and other tricks, 3DV-RPE is indeed helpful.**
>
> As answered in Q3, we have reported the baseline performance in Table 6 (1st row and 5th row) and attached the results for reference. Accordingly, we observe that the 3DV-RPE can boost the AP25 from 71.4 to 77.8 and AP50 from 47.6 to 66.0. We will explicitly add the above discussion in the final revision.

---

> ### Author Response · Authors · 2023-11-21
> **Response to Reviewer fSn3 (2/2)**
>
> > **Q5. Additional analysis. With position encoding being the center of this paper, I think it necessary to conduct ablation studies on other common formats of position encoding, such as: Absolute position encoding, in both the formats you proposed like Eqn. 3 and Eqn. 4, or common sin-cos position encoding.**
>
> Great point! First, as answered in Q2, we have reported the comparisons with two advanced RPE schemes specially designed for the 3D object detection task, including the CRPE (Stratified Transformer) and CRPE (EQNet). Second, we have also conducted the ablation experiments on the mentioned absolute position encoding in both the formats you proposed, like Eqn. 3 (in the updated PDF Eqn. 5) and Eqn. 4 (in the updated PDF Eqn. 2).
>
> | method                                                        | epoch | AP25 | AP50 |
> | ------------------------------------------------------------- | ----- | ---- | ---- |
> | Baseline (w/o RPE)                                            | 540x   | 71.4 | 47.6 |
> | Baseline + APE w/ Sin-Cos                                     | 540x   | 71.8 | 47.9 |
> | Baseline + APE w/ MLP + NonLinear（Eqn. 4, update to Eqn. 2）                    | 540x   | 72.1 | 48.7 |
> | Baseline + APE w/ MLP + NonLinear + Predefined Table（Eqn. 3, update to Eqn. 5 ） | 540x   | 72.0 | 48.5 |
> | Baseline + 3DV-RPE                                             | 540x   | 77.8 | 66.0 |
>
> According to the above results, we can observe that
>
> - Our 3DV-RPE significantly outperforms all the APE variants by a large margin, especially on AP50 which requires more accurate localization capability.
> - By comparing the 3rd row to the 2nd row, we observe that the MLP and non-linear transformation can slightly improve the performance of APE.
>
> > **Q6. More justifications of hyper-parameters. For example, where does 10 come from in $\mathbf{T}$'s shape? May I use another number to replace 10?**
>
> Good point! We chose $10$ considering the trade-off between performance and memory cost. We conducted ablation experiments on the size of $\mathbf{T}$ and reported the detailed comparison results in Table 12. We also included the results below for reference:
>
> | 3DV-RPE table shape    | \#epoch | AP25 | AP50 |
> | ---------------------- | ------- | ---- | ---- |
> | $5\times5\times5$    | 360x    | 76.7 | 64.7 |
> | $10\times10\times10$ | 360x    | 76.7 | 65.0 |
> | $25\times25\times25$ | 360x    | 76.7 | 64.2 |
>
> Accordingly, we have observed that our 3DV-RPE remains robust when faced with different choices for the shape of $\mathbf{T}$.

---

> ### Comment · Reviewer_fSn3 · 2023-11-22
>
> Thanks for your response!
>
> Your answers address my concerns. It looks to me like most of my questions were caused by the presentation issue. Still, I suggest the authors make (1) Table 6 more explicit, maybe adding a "(Baseline)" after None; (2) mentioning Table 13 somewhere in the main paper, I personally think it very important.
>
> I worked on 3D detection 1-2 years ago so I may not be the right person to justify the novelty as TFsP or YhUd. I tend to accept this paper for now, and will be willing to increase my score if the authors can convince them of the technical novelty.

---

> ### Author Response · Authors · 2023-11-23
> **Thanks for the Response of Reviewer fSn3**
>
> We would like to extend our sincere appreciation to the reviewer for your swift response and beneficial suggestions aimed at enhancing our presentation.
>
> Pursuant to your additional recommendations, we have implemented the following changes:
>
> - To enhance clarity, we replaced "None" with "Our Baseline" in Table 6 in the latest PDF revision.
>
> - We would like to apologize for the oversight in our previous response to Q2, where we failed to specify that Table 13 can be found on page 13.
>
> We are grateful for your professional integrity and research expertise. We've greatly benefited from your meticulous reviews. With respect to the technical novelty of our work, we kindly invite you to review the comments from Reviewer c9AZ, and our supplementary response regarding the additional inference advantage of our methodology compared to the previous SOTA systems.
>
> Once again, thank you for offering such invaluable feedback and for your support in accepting this paper.🤗

---

### Official Review · Reviewer_TFsP · 2023-10-31

**Soundness:** 2 fair
**Presentation:** 2 fair
**Contribution:** 2 fair
**Rating:** 5
**Confidence:** 5

**Summary:**

For the task of indoor 3D object detecion from point clouds, previous sparse detectors neglects to tackle 3D queries outside the bounding box, so the author proposed a 3D vertex positional encoding module (3DV-RPE) to guarantee the locality principle in object detection. Specifically, the proposed method encode relative position information for each query towards its assigned / predicted bounding boxes to provide clear information to guide the model to focus on points near the objects. Moreover, it utilized many widely-adopted tricks to generally improve the performance of the detector (custom backbone / loss normalization / TTA / one-to-many auxillary loss, etc). 3DV-RPE shows competitive results on ScanNetv2 and SUN-RGBD compared to previous methods.

**Strengths:**

* this paper pinpoint a interesting gap in previous sparse 3D indoor detectors that how to deal with queries outside the predicted bouding box, and try to prove that whether it is benefical to take this into account for 3D object detection.
* this paper does a lot of work to incorporate modern network architectures (e.g., ResNet34 + FPN), training strategy improvements to make the detector performs better.

**Weaknesses:**

* Unfair comparision:
    1) The author combines many technical improvements including normalizing box according to object size, one-to-many assignment as auxillary loss, a modified resnet34-fpn backbone, and even TTA. They're all irrelavant to the claimed core contribution 3DV-RPE. So to prove the proposed module is effective, the most convincing way could be adding 3DV-RPE directly onto the baseline GroupFree3D.  Considering that the 3DV-RPE  module can work in a plug-and-play manner in theory, I would expect more results based on other methods such as GroupFree3D, 3DETR or CAGroupFree3D.
    2) in table 6, the author reports the one with 3DV-RPE + TTA (77.8 / 66.0), does all other ablation attention results are also reported with TTA?
    3) the author reported the best results for the proposed method, how about the ablations? how many times have you run for each ablation choice in all tables? Does the fairness of the comparison is guaranteed?
    4) Given the best set of paraters for the authors final model, change the choices in loss functions can affect the hungarian matching cost matrix, thus it's hard to say the improvement / performance drop comes from the module / improper cost weights.
    5) Why some ablations are done with ScanNet while others use SUN-RGBD (e.g., Table 4)? Does it means the coordinates normalization works similarly on ScanNetv2?

* minor contributions:
    1) Actually I think the RPE and normalized coords are designed in similar ways: Point-RCNN has adopted to convert box to  canonical coords and do normalizations on oritentaion. Moreover, in anchor-based detectors, they already use the anchor boxes' W and H to normalize the regression targets. Here the author uses dynamic bounding boxes from predictions, which has also been explored in methods like MetaAnchor, etc.
    2) So many un-relavent tricks to improve the detection performance. I don't like the way to do whatever it can to improve the results. Rather, the author should focus on the main contribution. After the core module being sufficiently discussed, one can further improve its results with more tricks. Here the author put all stuff together, which makes me doubt where the improvement come from.

* Most of the references are before year 2023, so I think more recent works in year 2023 should be included.

* I recommend against reporting results using TTA, as this leads to cutthroat competition and more potentially unfair comparisons; for example, TTA may be different in different papers, but is always written as "TTA".

**Questions:**

* How much fraction does the queries outside predicted bounding boxes account for with respect to the total number of queries? 10%? 20%? The author should provide a investigation to this problem.
* Why the non-linear functions is designed in this way? how it is derived? is their any insights to do so?
* Why does the PE is added in the way in Eq. 1? I think the form of matmul(Q, K) + R does not match the intended aim of the paper. Instead, I think matmul(Q+R, K) should be more proper? or add a relative PE to Q and a global PE to K?

---

> ### Author Response · Authors · 2023-11-21
> **Response to Reviewer TFsP (1/3)**
>
> We thank the reviewer for the careful reviews and constructive suggestions. Above all, **please refer to the general response regarding concerns about unfair comparisons and minor contributions**. We address the other questions as follows:
>
> > **Q1. The author combines many technical improvements including normalizing box according to object size, one-to-many assignment as auxillary loss, a modified resnet34-fpn backbone, and even TTA. They're all irrelavant to the claimed core contribution 3DV-RPE. So to prove the proposed module is effective, the most convincing way could be adding 3DV-RPE directly onto the baseline GroupFree3D. Considering that the 3DV-RPE module can work in a plug-and-play manner in theory, I would expect more results based on other methods such as GroupFree3D, 3DETR or CAGroupFree3D.**
>
> Please refer to the general response for more details. In summary, our 3DV-RPE can significantly enhance the performance of both 3DETR and GroupFree3D. We will include these detailed comparison results in the final revision.
>
> > **Q2. in table 6, the author reports the one with 3DV-RPE + TTA (77.8 / 66.0), does all other ablation attention results are also reported with TTA?**
>
> Yes, we report all the ablation attention experiment results with TTA in Table 6. Furthermore, we presume you might also be interested in **the comparison results without TTA**, for which we provide the details as follows:
>
> | method | # Epochs | AP25 | AP50 |
> | ---------------------------- | -------- | ---- | ---- |
> | None                         | 360x      | 67.9 | 43.5 |
> | 3D Box Mask                  | 360x      | 72.9 | 58.3 |
> | 3DV-RPE                      | 360x      | 76.2 | 64.2 |
> | 3D Box Mask + 3DV-RPE        | 360x      | 75.3 | 61.5 |
> | None                         | 540x      | 70.6 | 46.7 |
> | 3D Box Mask                  | 540x      | 74.2 | 59.6 |
> | 3DV-RPE                      | 540x      | 77.4 | 65.0 |
> | 3D Box Mask + 3DV-RPE        | 540x      | 76.5 | 62.4 |
>
> In conclusion, even without TTA, our 3DV-RPE continues to significantly enhance performance, especially the AP50 metrics, which necessitates precise 3D localization capabilities.
>
> > **Q3. the author reported the best results for the proposed method, how about the ablations? how many times have you run for each ablation choice in all tables? Does the fairness of the comparison is guaranteed?**
>
> Yes, we have reported the best results from the ablation experiments.
>
> Following the previous GroupFree3D methodology, we trained each setting $5\times$ times and tested each training trial $5\times$ times. We have reported the maximum performance from these $25\times$ trials to ensure fairness. Additionally, we are prepared to provide the average performance, if necessary.
>
> > **Q4. Given the best set of paraters for the authors final model, change the choices in loss functions can affect the hungarian matching cost matrix, thus it's hard to say the improvement / performance drop comes from the module / improper cost weights.**
>
> Good point! We attempt to address your concerns about the impact of the loss function and Hungarian matching cost choices by replacing both with the original versions used in 3DETR, and we've reported **the comparison results without using TTA**:
>
> | method | loss weights                                                                                                      | cost weights                                                                                                      | # Epochs | AP25 | AP50 |
> | ------ | ----------------------------------------------------------------------------------------------------------------- | ----------------------------------------------------------------------------------------------------------------- | -------- | ---- | ---- |
> | Ours   | $\lambda_{1}=2$,$\lambda_{2}=1$,$\lambda_{3}=0.5$,$\lambda_{4}=3$,$\lambda_{5}=0.1$,$\lambda_{6}=0.5$ | $\lambda_{1}=2$,$\lambda_{2}=1$,$\lambda_{3}=0.5$,$\lambda_{4}=3$,$\lambda_{5}=0.1$,$\lambda_{6}=0.5$ | 540x     | 77.4 | 65.0 |
> | 3DETR  | $\lambda_{1}=1$,$\lambda_{2}=5$,$\lambda_{3}=1$,$\lambda_{4}=2$,$\lambda_{5}=0.1$,$\lambda_{6}=0.5$   | $\lambda_{1}=2$,$\lambda_{2}=0$,$\lambda_{3}=0$,$\lambda_{4}=1$,$\lambda_{5}=0$,$\lambda_{6}=0$       | 540x     | 77.0 | 64.7 |
>
> According to the above comparison experiments, we can see that **the significant differences in the choices of loss functions and matching cost functions only bring slight performance gains**. We will include these results in the final revision.
>
> > **Q5. Why some ablations are done with ScanNet while others use SUN-RGBD (e.g., Table 4)? Does it means the coordinates normalization works similarly on ScanNetv2?**
>
> The primary reason is that the ground-truth rotation angle annotations for all 3D bounding boxes in ScanNet are set to zero. Hence, the normalization of coordinates does not change the relative coordinates. We will incorporate this information into the final revision.

---

> ### Author Response · Authors · 2023-11-21
> **Response to Reviewer TFsP (2/3)**
>
> > **Q6. Actually I think the RPE and normalized coords are designed in similar ways: Point-RCNN has adopted to convert box to canonical coords and do normalizations on oritentaion. Moreover, in anchor-based detectors, they already use the anchor boxes' W and H to normalize the regression targets. Here the author uses dynamic bounding boxes from predictions, which has also been explored in methods like MetaAnchor, etc.**
>
> Please refer to the general response. We will add references and discussions on connections to acknowledge related efforts in the final revision. We sincerely hope that our work will not be rejected solely because some aspects of our V-DETR share similar insights with previous studies.
>
> > **Q7. So many un-relavent tricks to improve the detection performance. I don't like the way to do whatever it can to improve the results. Rather, the author should focus on the main contribution. After the core module being sufficiently discussed, one can further improve its results with more tricks. Here the author put all stuff together, which makes me doubt where the improvement come from.**
>
> Good point! We would like to improve the organization of the experiments for clarity. Additionally, we have applied our 3DV-RPE to several other clean baselines, including 3DETR and GroupFree3D. Please refer to the detailed results in the general response
>
> > **Q8. Most of the references are before year 2023, so I think more recent works in year 2023 should be included.**
>
> Great point! We have carefully revisited the related works from 2023 and summarized a list of their results as follows. We would appreciate any additional valuable comments on the related works from 2023 that we may have missed. **We have added references and comparisons with these works in Table 1 of the revised PDF.**
>
> | results in ScannetV2    | AP25 | AP50 |
> | ----------------------- | ---- | ---- |
> | VDETR(no TTA)           | 77.4 | 65.0 |
> | Point-GCC * [1]         | 73.1 | 59.6 |
> | Uni3DETR [2]            | 71.7 | 58.3 |
> | AShapeFormer[3]         | 71.1 | 56.6 |
> | Swin3D(no pretrain) [4] | 73.3 | 58.6 |
> | Swin3D**                | 76.4 | 63.2 |
>
> | results in SUN RGB-D | AP25 | AP50 |
> | -------------------- | ---- | ---- |
> | VDETR(no TTA)        | 67.5 | 50.4 |
> | Point-GCC* [1]       | 67.7 | 51.0 |
> | Uni3DETR [2]         | 67.0 | 50.3 |
> | AShapeFormer[3]      | 62.2 | -    |
> | OctFormer [5]        | 66.2 | 50.6 |
> | ConDaFormer[6]       | 67.1 | 49.9 |
>
> *use self-supervised pretrainning
>
> **use Extra data pretrainning
>
> [1] Point-GCC: Universal Self-supervised 3D Scene Pre-training via Geometry-Color Contrast, arXiv 2023
>
> [2] Uni3DETR: Unified 3D Detection Transformer, NeurIPS 2023
>
> [3] AShapeFormer: Semantics-Guided Object-Level Active Shape Encoding for 3D Object Detection via Transformers, CVPR 2023
>
> [4] Swin3D: A Pretrained Transformer Backbone for 3D Indoor Scene Understanding, arXiv 2023
>
> [5] OctFormer: Octree-based Transformers for 3D Point Clouds, arXiv 2023
>
> [6] ConDaFormer: Disassembled Transformer with Local Structure Enhancement for 3D Point Cloud Understanding, NeurIPS 2023
>
> > **Q9. How much fraction does the queries outside predicted bounding boxes account for with respect to the total number of queries? 10%? 20%? The author should provide a investigation to this problem.**
>
> We guess you're referring to the fraction of queries localized outside the ground-truth bounding boxes, rather than within the predicted bounding box. Each query will predict a center and dimensions, ensuring that it is localized at the center position of the predicted bounding box.
>
> We compile statistics on the fraction of queries that fall outside the predicted bounding boxes in relation to all queries as follows:
>
> |                                                           | \#8-th decoder layer | \#6-th decoder layer | \#4-th decoder layer | \#2-th decoder layer | first-stage |
> | --------------------------------------------------------- | -------------------- | -------------------- | -------------------- | -------------------- | ----------- |
> | \# of the queries outside ground-truth box/\# all queries | 24.77%               | 25.62%               | 27.53%               | 32.79%               | 38.21%      |
>
> In addition, we also compile statistics on the fraction of queries localized outside the ground-truth bounding boxes, considering only the matched queries selected by Hungarian matching as follows:
>
> |                                                                       | \#8-th decoder layer | \#6-th decoder layer | \#4-th decoder layer | \#2-th decoder layer | first-stage |
> | --------------------------------------------------------------------- | -------------------- | -------------------- | -------------------- | -------------------- | ----------- |
> | \# of the matched queries outside ground-truth box/\# matched queries | 1.14%                | 1.77%                | 2.64%                | 4.26%                | 5.92%       |

---

> ### Author Response · Authors · 2023-11-21
> **Response to Reviewer TFsP (3/3)**
>
> > **Q10. Why the non-linear functions is designed in this way? how it is derived? is their any insights to do so?**
>
> We have reported the detailed comparison results of different non-linear functions in Table 2. Accordingly, we observe consistent gains with different non-linear functions.
>
> The key insight is that, due to the points being primarily distributed on the surface of the 3D object, the points near the surface of the object tend to hold the most important semantic information. This information can determine the category and size of a prediction box. Therefore, **relative position encoding should be more sensitive to smaller spatial changes than to larger changes in the 3D point cloud space**.
>
> Therefore, **the non-linear function should be capable of magnifying small changes in relative coordinates**. We will include this information in the final revision.
>
> > **Q11. Why does the PE is added in the way in Eq. 1? I think the form of matmul(Q, K) + R does not match the intended aim of the paper. Instead, I think matmul(Q+R, K) should be more proper? or add a relative PE to Q and a global PE to K?**
>
> We address your concerns on the PE scheme as follows:
>
> - First, we need to clarify that relative position encoding should explicitly encode the relative spatial relationships between each query and each key. Therefore, by default, the shape of the RPE should be $nQ\times nK$. Our implementation that directly adds RPE to matmul(Q, K) is the most intuitive one.
> - Second,  the motivation for adding the RPE to matmul(Q, K) is to explicitly modulate the pairwise cross-attention map values between the 3D bounding box query and each of the 3D point clouds. In other words, the key (and value) point clouds localized in the query 3D boxes will receive larger values and smaller values otherwise.
>
> Therefore, simply adding the PE to a query or the key essentially results in absolute PE instead of relative PE. We also present the results **(w/o TTA)** with absolute PE, which adds PE to both query and key, in the following table. As you can see, the absolute PE does not perform well.
>
> | method                  | epoch | AP25 | AP50 |
> | ----------------------- | ----- | ---- | ---- |
> | Baseline + abosolute PE | 540x   | 70.7 | 47.1 |
> | Baseline + 3DV-RPE      | 540x   | 77.4 | 65.0 |
>
> In summary, the proposed RPE scheme provides a natural and effective implementation. We will include the above information in the final revision if necessary.
>
> We would also like to invite the reviewer to take another look at an insightful study[1] on the relative position encoding scheme for Vision Transformers. Their experiments indicate superior performance of both the bias and contextual modes. **Our 3DV-RPE can be considered a effective designed version of the bias mode, specifically tailored for 3D object detection tasks.**
>
> [1] Rethinking and Improving Relative Position Encoding for Vision Transformer, ICCV 2021

---

> ### Comment · Reviewer_TFsP · 2023-11-22
> **Final**
>
> Thank authors for the detailed responses.
>
> After reviewing the rebuttal and the revised paper, it still cannot convince me on questions Q4, Q6, Q7 and Q9. Besides, the novelty of this paper is not enough to me: similar operations has been introduced in previous methods.
>
> I suggest the authors to re-orgainze the paper to foucs more on the main contribution, and do more work to make it distinct from previous methods.
>
> I strongly recommend that authors remove TTA and some very tricky operations, because this will cause other authors to have to spend more effort on doing everything possible to improve performance, rather than seek for what really matters.
>
> As a result, I'd like to keep my score unchanged.

---

> ### Author Response · Authors · 2023-11-22
> **Thanks for the Response of Reviewer TFsP**
>
> We are grateful for the reviewer's swift response. 🤗
>
> 👉 First, we are seeking further clarification on the "similar operations" introduced in previous methods. As highlighted in our general response, the fundamental contribution of our work is the novel 3DV-RPE and its efficient variant, which has significantly enhanced the AP50 performance across an array of baselines. **To the best of our knowledge, the 3DV-RPE scheme, which explicitly models the spatial relations between arbitrary points and the eight vertices of a 3D bounding box, is an innovative approach unexplored in prior studies.**
>
> 👉 Second, we welcome any specific suggestions that could assist us in addressing questions Q4, Q6, Q7, and Q9 more effectively.
>
> 👉 Last, we would like to emphasize that **our V-DETR maintains superior performance even without the application of TTA**. Other operations are primarily inspired by well-established practices in the DETR-based 2D object detection systems. We are committed to releasing the source code, aiming to aid other researchers in focusing their efforts on elements of true significance.
>
> We greatly appreciate your invaluable and insightful comments once again!🤗

---

### Official Review · Reviewer_c9AZ · 2023-10-31

**Soundness:** 3 good
**Presentation:** 4 excellent
**Contribution:** 4 excellent
**Rating:** 8
**Confidence:** 4

**Summary:**

The manuscript proposes a 3d object-detection-specific position encoding method that significantly improves performance for 3d DETR-like object detection.
The 3d position encoding encodes the relative position of key, value points to the object query points to allow the transformer to learn to attend to points inside the object bounding box more easily.
With the addition of the position encoding and some other tweaks, the proposed method, V-DETR, outperforms CNN-based methods. A first for transformer-based detectors in 3d.

**Strengths:**

The proposed approach for relative position encoding is intuitive (and illustrated well in Fig 1), and experiments clearly show that it leads to a big improvement for Transformer-based methods and leads to a new state of the art wrt. to CNN-based methods as well.

Overall the quality of writing and illustration is very high. The detailed pipeline visualization clearly shows the recurrent nature of the approach. The visualization of the attention for each of the corners is also very illustrating.

The manuscript pays attention to practical aspects as well: The use of a precomputed lookup table for the relative PE is a nice and practical way to safe valuable GPU memory.

The experiments are expansive and convincing. The ablations do help clarify the different choices of the hyperparameters.

**Weaknesses:**

The precomputed lookup table was the hardest to follow (Eq 3) since the connection to Eq 4 was not immediately obvious. One more sentence there to explicitly connect the two would be helpful. I.e. T represents a discretized set of possible \Delta P that we interpolate into.

**Questions:**

Page 5 has a broken figure reference.

I dont understand how T in Eq(3) is initialized/set? What range do the T values take? -5 to 5 as indicated by the signed-log function?

---

> ### Author Response · Authors · 2023-11-21
> **Response to Reviewer c9AZ**
>
> We thank the reviewer for the careful reviews and constructive suggestions. We answer the questions as follows.
>
> > **Q1. The precomputed lookup table was the hardest to follow (Eq 3) since the connection to Eq 4 was not immediately obvious. One more sentence there to explicitly connect the two would be helpful. I.e. T represents a discretized set of possible \Delta P that we interpolate into.**
>
> Thanks for pointing out this issue and providing great suggestions! **We have followed your suggestions to add the mentioned sentence between Eq. 3 and Eq. 4 (in the updated PDF: between Eq. 2 and Eq. 5) to make it clearer in the revised PDF.**
>
> Additionally, following the comments from Reviewer fSn3, we've noted that the explanation could be further clarified by discussing "3DV-RPE" prior to introducing the concept of the "canonical object space".
>
> > **Q2. Page 5 has a broken figure reference.**
>
> Thanks for pointing out this issue! We will fix the typo "fig:rotatedRPE" (refers to Figure 4) in the final revision.
>
> > **Q3. I dont understand how T in Eq(3) is initialized/set? What range do the T values take? -5 to 5 as indicated by the signed-log function?**
>
> 👉 The $\mathbf{T}$ is initialized/set with a 3D grid of size $10\times10\times10$. Each value is a 3-d vector and the values at position $(i,j,k)$ is computed by $\mathbf{T}[i, j, k] = (\frac{\mathrm{max value}(2i-(\mathrm{table size}-1))}{\mathrm{tablesize}-1},\frac{\mathrm{maxvalue}(2j-(\mathrm{tablesize}-1))}{\mathrm{tablesize}-1},\frac{\mathrm{maxvalue}(2k-(\mathrm{tablesize}-1))}{\mathrm{tablesize}-1})$.  (Eq(3) has been reordered to Eq(5) in the updated PDF)
>
> 👉 Yes, the $\mathbf{T}$ values are within the range of -5 to 5. We also provide a concise python implementation of the $\mathbf{T}$ initialization as follows:
>
> ```python
> MAX_VALUE = 5
> TABLE_SIZE = 10
>
> # the initialization of T
> relative_coords_table = torch.stack(torch.meshgrid(
>     torch.linspace(-MAX_VALUE, MAX_VALUE, TABLE_SIZE, dtype=torch.float32),
>     torch.linspace(-MAX_VALUE, MAX_VALUE, TABLE_SIZE, dtype=torch.float32),
>     torch.linspace(-MAX_VALUE, MAX_VALUE, TABLE_SIZE, dtype=torch.float32),
> ), dim=-1).unsqueeze(0)
>
> self.register_buffer("relative_coords_table", relative_coords_table)
> ```
>
> We create and initialize $\mathbf{T}$ with the meshgrid function and linspace function in pytorch. **We would like to add the above source code of the efficient implementation of 3DV-RPE in the final revision to make it clearer. We also ensure to release the source code of our approach after the paper is accepted.**

---

> > ### Comment · Reviewer_c9AZ · 2023-11-22
> >
> > Thanks for the clarifications by the Authors. They address my questions well.
> > I have also read the reviews of the other reviewers. I think even though the local coordinate system encoding may have been used in other works, it has not been used and shown effective for 3D-DETR.
> > The modification to the original DETR may be "small" but the effect is significant. To me this is actually an advantage rather than a reason for reject.
> > Ill stick with accept for the paper.

---

> > > ### Author Response · Authors · 2023-11-23
> > > **Thanks for the Response of Reviewer c9AZ**
> > >
> > > We express our gratitude to the reviewer for the expedient response and for the sustained support reflected in the positive rating.🤗
> > >
> > > We will integrate the contents of the rebuttal into our final revision, guided by your invaluable suggestions. Furthermore, we commit to releasing the source code of our methodology once the paper is accepted. Our hope is that this will contribute to and benefit the indoor 3D object detection community along the path of DETR-based approaches.

---

### Official Review · Reviewer_YhUd · 2023-11-01

**Soundness:** 3 good
**Presentation:** 2 fair
**Contribution:** 2 fair
**Rating:** 5
**Confidence:** 4

**Summary:**

This paper presents one improvement over DETR-based methods for 3D object detection. The idea is first to predict a coarse bounding box and then only allow attention weights to be learned within the bounding box for better-using locality as one important inductive bias for 3D object detection. Experiments are done on the ScanNetV2 and Sun RGB-D benchmarks. Results show improved performance over baselines. Extensive ablation studies are also presented.

**Strengths:**

- the proposed method improves over state-of-the-art methods on ScanNetV2 and Sun RGB-D.
- the proposed modification to the DETR-based method is valid and reasonable.
- ablation studies are solid.

**Weaknesses:**

- the proposed method is more like just a small fix to DETR-based method.
- the proposed fix is also specific to DETR-based backbones.
- the proposed fix may be vulnerable if the first stage predicting the coarse bounding boxes fail. For example, if the bounding boxes are very off, then preventing later layers to attend to out-of-the-box regions may make it impossible to recover.
- the paper writing can be improved. For example, the figure layouts are quite messy. The organization for Sec. 3.1 and 3.2 is a bit hard to follow. It looks like Sec. 3.1 focuses on laying out the basic pipeline of DETR and Sec. 3.2 discusses more into the contributions of the paper, but actually the content are mixed together.
-  there are also claims that are unsupported in the paper. For example, the sentence in the introduction section "We attribute the discrepancy to the limited scale of training data available for 3D object detection" is not well supported. Can you use less data to train 2D detectors to show it's really the data scale issue?

**Questions:**

see weakness

---

> ### Author Response · Authors · 2023-11-21
> **Response to Reviewer YhUd**
>
> We thank the reviewer for the careful reviews and constructive suggestions. We answer the questions as follows.
>
> > **Q1. the proposed method is more like just a small fix to DETR-based method.**
>
> Refer to the general response.
>
> > **Q2. the proposed fix is also specific to DETR-based backbones.**
>
> Good point! We want to point out that DETR-based methods have dominated the SOTA in 2D detection and 3D outdoor detection. Therefore, the proposed fix shows that the 3DV-RPE is key to building the SOTA 3D indoor detection system with the DETR-based method. We would also like to explore how to apply the proposed method to non-DETR-based methods in the future, based on your further valuable comments.
>
>
> > **Q3. the proposed fix may be vulnerable if the first stage predicting the coarse bounding boxes fail. For example, if the bounding boxes are very off, then preventing later layers to attend to out-of-the-box regions may make it impossible to recover.**
>
> Good point! However, we have empirically found that this concern might not be valid, as we explain in detail in the following response.
>
> First, we have compiled statistics on the ratio of (matched) first-stage predictions that fall outside the ground-truth bounding boxes. We found that **only 4.35% of the matched proposal boxes have their center points outside the target box, and a mere 0.06% have zero intersection over union (IoU) with the target box at all**. This is a relatively rare occurrence. **The reason is that DETR-based methods apply the Hungarian matching technique to select the top-ranking high-quality predictions based on cost functions that measure the accuracy of bounding box localization.** Therefore, the (matched) first-stage predictions are generally accurate enough to be used as input for the second stage.
>
> Second, we have also compiled statistics on the ratio of (matched) predictions after refinement through subsequent transformer decoder layers. The proportion of refined predictions with their center points outside the target box decreases from 4.35% to 1.14%, and those with zero IoU at all drop to 0.02%. This indicates that our 3DV-RPE is capable of recovering low-quality first-stage predictions. The key reason is that **our 3DV-RPE can also capture useful long-range contextual information outside the box.** We have conducted a detailed analysis in the Section "Comparison with 3D box mask" (page 7) and in Table 6 (page 9).
>
> We will conduct further studies if you can provide additional valuable comments.
>
> > **Q4. the paper writing can be improved. For example, the figure layouts are quite messy. The organization for Sec. 3.1 and 3.2 is a bit hard to follow. It looks like Sec. 3.1 focuses on laying out the basic pipeline of DETR and Sec. 3.2 discusses more into the contributions of the paper, but actually the content are mixed together.**
>
> Great suggestion! We would like to improve the layout of the figures and restructure the organization of Sec. 3.1 and Sec. 3.2 to make it clearer.
>
> > **Q5. there are also claims that are unsupported in the paper. For example, the sentence in the introduction section "We attribute the discrepancy to the limited scale of training data available for 3D object detection" is not well supported. Can you use less data to train 2D detectors to show it's really the data scale issue?**
>
> Thanks for pointing out this issue! We agree this claim is not well supported and we would like to clarify it as follows.
>
> 👉 Our primary point is that **the scale of 3D object detection data is significantly smaller compared to that of 2D object detection tasks.** For instance, the number of different scenes (representing valid training samples) in ScanNetv2 is only 1,513, while the number of different images in COCO exceeds 110,000, rendering it approximately 70 times larger. We hypothesize that the limited amount of training data may complicate the learning of inductive biases, including locality.
>
> 👉 Furthermore, we conduct experiments on the 2D detector DETR, using approximately 1% of the training data (1,200 images), following your suggestion. We train DETR[1] for the same number of iterations (2,217,881 iterations) as the original DETR, which was trained on the full dataset for 300 epochs, while maintaining a batch size of 16. First, we observe that the mAP of the validation set drops from 44.9% to 10.4%, which closely aligns with the performance reported in Table 5 of [2]. Second, in accordance with Figure 6 of the DETR paper, **we visualize the cross-attention maps for the predicted object in Figure 11 of the revised PDF**. **We note that these attention maps fail to focus on local object regions, especially the object extremities. Therefore, it is evident that the scale of data has a substantial impact on the model's ability to effectively learn the locality inductive biases.**
>
> [1] End-to-End Object Detection with Transformers, ECCV 2020
>
> [2] DETReg: Unsupervised Pretraining with Region Priors for Object Detection, CVPR2022

---

### Author Response · Authors · 2023-11-21
**General Response to All Reviewers and AC**

We extend our deepest gratitude to all the reviewers for their meticulous reviews and constructive suggestions.🤗

We first address the major concerns raised about unfair comparison and minor contribution:

> **unfair comparision (TFsP)**

👉 First, we need to clarify that **all the ablation experiments are conducted under fair settings**. For example, in Table 6, we have shown that applying the proposed 3DV-RPE improves our enhanced baseline (which combines all other improvements) by +6.4 on AP25 and +18.4 on AP50. This is a significant gain achieved under fully fair comparison settings. Second, for the system-level comparisons reported in Table 1, we must admit that it is challenging to ensure fully fair comparisons since different SOTA systems have employed various complex additional tricks. For instance, CAGroup3D uses an improved BiResNet backbone that brings obvious gains on AP25 metrics.

👉 Second, following your suggestions, we have implemented our method over the official code of 3DETR and GroupFree3D. We provide the detailed comparison results on ScanNet under fair settings as follows:

|                 | \#epoch | AP25        | AP50         |
| --------------- | ------- | ----------- | ------------ |
| 3DETR           | 1080x   | 65.0        | 47.0         |
| 3DETR + 3D-VPRE | 1080x   | 71.2 (+6.2) | 60.8 (+13.8) |

|                       | \#epoch | AP25        | AP50        |
| --------------------- | ------- | ----------- | ----------- |
| GroupFree3D           | 400x    | 69.1        | 52.8        |
| GroupFree3D + 3D-VPRE | 400x    | 72.8 (+3.7) | 62.1 (+9.3) |

We set the number of training epochs following their official implementation [1][2]. Based on these results, we can draw a similar conclusion that **our proposed 3DV-RPE significantly enhances both 3DETR and GroupFree3D**. These results will be included in the final revision.

[1] https://github.com/facebookresearch/3detr

[2] https://github.com/zeliu98/Group-Free-3D

> **minor contributions (TFsP), a small fix to DETR-based method (YhUd).**

👉 First and foremost, we wish to clarify that the primary motivation of this work is the following observation: while DETR-based methods have achieved state-of-the-art results in 2D detection[1] and 3D outdoor detection[2], they significantly lag behind in 3D indoor detection. **The objective of this work is to bridge this substantial gap and provide a strong baseline using a DETR-based method. We consider this to be a noteworthy contribution to the community**. We also commit to releasing the source code of our implementation once the paper is accepted.

[1] DETRs with Collaborative Hybrid Assignments Training, ICCV 2023

[2] Far3D: Expanding the Horizon for Surround-view 3D Object Detection, arXiv 2023

👉 Second, the primary technical contribution of this work lies within the 3DV-RPE. This technique is non-trivial and significant for two reasons:

1. The fundamental challenge in 3D point cloud recognition tasks is the sparsity and irregularity. Transferring the success of powerful 2D operators such as the multi-scale Deformable Attention operations introduced by Deformable-DETR[1] to 3D point clouds is a daunting task. Our 3DV-RPE aims to model the accurate spatial relations in 3D space, and we empirically demonstrate its effectiveness, particularly in terms of significant gains on the AP50 metrics, e.g., 3DV-RPE brings +18.4, +13.8, and +9.3 gains over our baseline, 3DETR, and GroupFree3D, respectively.

[1] Deformable DETR: Deformable Transformers for End-to-End Object Detection, ICLR 2021

2. To the best of our knowledge, we are the first to explicitly model the spatial relations between each point and the eight vertices of a 3D bounding box. As shown in Table 3, utilizing eight vertices yields a more than +10 improvement on the AP50 metrics compared to using one vertex. Furthermore, we also provide a highly efficient implementation of the 3DV-RPE through the use of a 3DV-RPE table. We believe this is a novel and effective technique that can be easily implemented in other 3D point cloud tasks.

In summary, **considering the precise 3D spatial modeling capability and the significant performance gains, we believe that our 3DV-RPE is a non-trivial contribution to the 3D object detection community.**

👉 Third, we would like thanks for the valuable comments on mentioning the related efforts including Point-RCNN and MetaAnchor (TFsP). We agree that the insignt of is similar with these mentioned work to some degree. We have added reference and the connection/difference discussions to acknowledge the related efforts in the final revision. We also want to highlight that the key innovation is at the 3DV-RPE and we sincerely hope our work won't be rejected solely based on some parts of our V-DETR sharing similar insights with previous studies.

👉 In summary, we believe our work is more than just a small fix to DETR-based method.

---

> ### Author Response · Authors · 2023-11-21
> **Summary of the Modifications made in the PDF Submission**
>
> We have made several changes to the PDF, with the modifications clearly highlighted in blue. Here is a summary of the revisions:
>
> 1. The 3DV-RPE section has been moved to an earlier position within Section 3.2.
>
> 2. We have inserted a discussion regarding the connection and difference with Point RCNN in the Canonical Object Spaces section.
>
> 3. In Table 1, we have added comparisons with the most recent methods from 2023, and included a related discussion in Section 4.3.
>
> 4. The Comparison with 3D box mask section has been relocated to a prior position in Section 4.4 to clarify the baseline results.
>
> 5. In Section C of the appendices, we have added the necessary ablation study focusing on the impact of data scale on the learning of locality inductive bias for 2D object detection.
>
> 👉 Due to the aforementioned modification, the equation numbers have been altered. Their correspondence is listed below:
>
> |     Original Equation Number     | Eq (1) | Eq (2) | Eq (3) | Eq (4) | Eq (5) |
> | ------------------------------ | -------------------- | -------------------- | -------------------- | -------------------- | ----------- |
> |     Updated Equation Number      | Eq (1) | Eq (4) | Eq (5) | Eq (2) | Eq (3) |

---

> ### Author Response · Authors · 2023-11-22
> **General Response to All Reviewers and AC**
>
> We would like to express our gratitude to all the reviewers for their invaluable feedback, which will undoubtedly assist us in significantly enhancing this submission.🤗🤗🤗
>
> 👉 First, we would also like to draw the reviewers' attention to the inference efficiency of our method as follows (refer more details on Table 14 on page 13 and the 'Inference Complexity Comparison' section on page 14):
>
>
> |          method  |  # Scenes/second |  Latency/scene |GPU Memory |
> | ---------------- | ---- | ---- | ---- |
> FCAF3D |7.8 | 128ms | 628M |
> CAGroup3D | 2.1 | 480ms | 1138M |
> Ours (light) |  7.7 |  130ms |  489M |
> Ours|  4.2 |  240ms |  642M |
>
> According to the above results, we can observe that our V-DETR system demonstrates a nearly $2\times$ speed improvement during inference while achieving better performance when compared to CAGroup3D. Additionally, we offer a lighter version by reducing the number of 3D object queries from $1024$ to $256$.
>
> 👉 Second, we emphasize that the previous state-of-the-art system, CAGroup3D, relies heavily on advanced architectural designs such as RoI-Conv (based on RoI pooling) and Voxel-wise Semantic and Vote Prediction (based on VoteNet). These designs demand substantial expertise in conventional 2D object detection and are not straightforward to implement.
>
> One of the reasons we used additional strategies to enhance the 3DETR is its simplicity and naivety when compared to CAGroup3D and recent state-of-the-art 2D object detection systems like DINO-DETR[1] and H-DETR[2]. To close the significant gap between the existing state-of-the-art DETR-based methods on indoor 3D tasks (on the AP25 metric, GroupFree3D and 3DETR trail behind CAGroup3D by 6 and 10 points respectively), we have exerted considerable effort to optimize the entire 3D object detection system from various aspects, excluding the primary contribution of the 3DV-RPE.
>
> In conclusion, we hope that our efforts could be recognized by all reviewers. Furthermore, we believe that the application of our V-DETR will also benefit the 3D object detection community.
>
> [1] DINO: DETR with Improved DeNoising Anchor Boxes for End-to-End Object Detection, ICLR 2023
>
> [2] DETRs with Hybrid Matching, CVPR 2023

---

### Meta-Review · Area_Chair_ZAuq · 2023-12-09

**Metareview:**

**Summary**

The paper proposes to improve 3D object detection with DETR-based backbones by introducing a relative position encoding (3DV-RPE) for points based on offsets to predicted bounding box vertices.  The relative positions are computed based on the canonical coordinates for the object.  Experiments and ablations show the proposed modifiation results in significant improvement in 3D object performance.


**Strengths**
- The proposed relative encoding, while sharing some similarities with prior work, has some differences and has not been applied to DETR-based backbones for 3D detection before.
- The proposed method works well, with a expansive set of experiments and ablations that show it can outperform prior work
- The work pays attention to practical aspects of implementing the model efficiently

**Weaknesses**
- The proposed method is very specific to DETR-based backbones as it is a small modification on top of DETR (YhUd) and is similar to relative position encoding used in Point-RCNN (TFsP)
- Paper organization and presentation could be improved (YhUd)

**Recommendation**
- The AC recommends the paper be accepted as a poster as the proposed method is effective, but the relative position encoding is a fairly small modification to 3D-DETR.

**Suggested improvements to paper**
1. Add experiments 1) showing the usefulness of adding 3DV-PRE to 3DETR and GroupFree3D (from general response) to main paper and 2) comparison of 3DV-PRE with alternatives without TTA (from response to TFsP) to Table 6.
  These (together with the existing Table 6) is the key experiment showing the usefulness of the proposed 3DV-PRE and should be emphasized more.
  The caption / header of Table 6 is actually confusing.  The AC is not sure why it is indicated as "attention modulation" choices.  Is this not the table that compares the different positional encodings?
2. Improve results layout and discussion.
  - Again, Table 6 should be improved to be with/without TTA + ablations comparing 3DV-PRE should be highlighted as the main results of this work (possibly moved ahead of comparisons with SOTA methods in Table 1)
  - The SOTA comparisons Table 1 can probably be reduced in size in the main paper - some results can be in the appendix
  - Tables should be reordered and repositioned so they appear in the order that they are referenced (e.g. the current Table 6 should be moved up)
3. Add additional experiments and clarification that was included as part of the author response (mostly in response to TFsP and fSn3)
4. Add information about efficient implementation of 3DV-RPE to appendix (response to c9AZ)

**Justification For Why Not Higher Score:**

- As pointed out by reviewers, the proposed relative position encoding is a fairly small modification on 3D DETR-based object detection and the main contribution of the work is the set of experiments showing the effectiveness of the encoding for 3D object detection on a DETR-backbone.  Thus the usefulness and impact of the work is limited.

**Justification For Why Not Lower Score:**

- Despite the fact that the modification is minor, it is effective and could be of interest to the 3D object detection community.

---

### Decision · Program_Chairs · 2024-01-16

Accept (poster)